Probabilistic model based on circular statistics for quantifying coverage depth dynamics originating from DNA replication

Suzuki Shinya
Yamada Takuji takuji@bio.titech.ac.jp
School of Life Science and Technology, Tokyo Institute of Technology , Meguro , Tokyo , Japan
Tatarinova Tatiana
Electronic publication date: 2020 Mar 27
Publication date: 2020
Volume: 8
Electronic Location ID: e8722
Received 2019 Oct 22; Accepted 2020 Feb 10
Copyright: ©2020 Suzuki and Yamada
Copyright year: 2020
Copyright holder: Suzuki and Yamada
License: This is an open access article distributed under the terms of the Creative Commons Attribution License, which permits unrestricted use, distribution, reproduction and adaptation in any medium and for any purpose provided that it is properly attributed. For attribution, the original author(s), title, publication source (PeerJ) and either DOI or URL of the article must be cited.
License URL: https://creativecommons.org/licenses/by/4.0/

Keywords: Growth rate estimation by metagenome sequence, Coverage depth, DNA replication model, Von mises generalized linear model, Peak to trough ratio, Metagenomics, Microbiome

Funding: JSPS KAKENHI 17J10014 AIP acceleration Research (JST) JPMJCR19U3 Japan Agency for Medical Research and Development (AMED) JP19cm0106464 ROIS National Institute of Genetics This work was supported by JSPS KAKENHI Grant Number 17J10014, AIP acceleration Research (JST) Grant Number JPMJCR19U3, and the Japan Agency for Medical Research and Development (AMED) Grant Number JP19cm0106464. A portion of the computations waw performed on the NIG supercomputer at ROIS National Institute of Genetics. The funders had no role in study design, data collection and analysis, decision to publish, or preparation of the manuscript.

==============================
Background

With the development of DNA sequencing technology, static omics profiling in microbial communities, such as taxonomic and functional gene composition determination, has become possible. Additionally, the recently proposed in situ growth rate estimation method allows the applicable range of current comparative metagenomics to be extended to dynamic profiling. However, with this method, the applicable target range is presently limited. Furthermore, the characteristics of coverage depth during replication have not been sufficiently investigated.

Results

We developed a probabilistic model that mimics coverage depth dynamics. This statistical model explains the bias that occurs in the coverage depth due to DNA replication and errors that arise from coverage depth observation. Although our method requires a complete genome sequence, it involves a stable to low coverage depth (>0.01×). We also evaluated the estimation using real whole-genome sequence datasets and reproduced the growth dynamics observed in previous studies. By utilizing a circular distribution in the model, our method facilitates the quantification of unmeasured coverage depth features, including peakedness, skewness, and degree of density, around the replication origin. When we applied the model to time-series culture samples, the skewness parameter, which indicates the asymmetry, was stable over time; however, the peakedness and degree of density parameters, which indicate the concentration level at the replication origin, changed dynamically. Furthermore, we demonstrated the activity measurement of multiple replication origins in a single chromosome.

Conclusions

We devised a novel framework for quantifying coverage depth dynamics. Our study is expected to serve as a basis for replication activity estimation from a broader perspective using the statistical model.

Introduction

The development of high-throughput DNA sequencers has enabled massive and exhaustive microbiome analyses. By mapping fragmented reads onto databases, the taxonomic and functional gene composition of a sample can be measured. Several researchers have utilized this procedure to investigate samples from various environments, such as those of human and animal bodies as well as those of other types of environmental samples (Hildebrand et al., 2013; Kato et al., 2015; Zhu et al., 2015; Higashi et al., 2018). One possible means to progressing beyond the profiling of static information involves investigating microbial dynamics. Although time-series microbiome profiling via quantitative polymerase chain reaction (PCR) or cell-sorting may allow dynamics measurement, such methods do not easily provide a comprehensive understanding of growth dynamics from the single sample involved therein (Tourlousse et al., 2017; Vandeputte et al., 2017). Meanwhile, the peak-to-trough ratio (PTR) of the coverage of whole genome sequencing (WGS) reads mapped to a reference genome sequence provides an estimate of growth; notably, this approach uses WGS reads from just a single sample (Korem et al., 2015). This method is based on the considerable increase in DNA around the replication origin via bidirectional DNA replication (Cooper & Helmstetter, 1968; Bremer & Churchward, 1977). As quantitative pipelines continuously undergo re-evaluation and extension, a draft quality genome sequence may be applied. Few methods have been proposed to quantify the growth of bacteria from genomic data. iRep uses a mechanism in which the slope of the sorted coverage on contig sequences was correlated with the growth rate (Brown et al., 2016). GRiD enables more robust estimation by sorting the coverage depths of multiple contigs (Emiola & Oh, 2018). DEMIC performed accurate estimation by using the coverage depths of multiple samples and estimating the appropriate position via principal component analysis (Gao & Li, 2018). Some studies using such pipelines have revealed associations between growth estimates and factors such as disease, 24-hour oscillations, and diet (Olm et al., 2017; Forsyth et al., 2018). Thus, such an approach quantifying the growth of bacteria from coverage depth is expected to facilitate the investigation of new fields of microbial research. However, some questions associated with coverage depth modeling remain unresolved.

The first challenge regarding growth estimation from coverage depth is related to the application scope of the method. Previous studies have enabled growth estimation for a broad range of samples, but the range of applicability remains limited. Taking coverage depth as an example, even the most robust method currently requires 0. 05 × average coverage with a complete sequence or 0. 2 × with a de novo-assembled sequence. A novel growth rate estimation method that is less sensitive to decreases in coverage depth could be utilized in broader applications. Second, the previously proposed pipelines are not applicable to microbes with multiple replication origins as these pipelines use a model based on a single peak and trough. This approach narrows the range of measurement targets as some taxa such as archaea have two or more replication origins in a single replicon (Lundgren et al., 2004; Robinson et al., 2004; Andersson et al., 2010). It has also been suggested that some bacteria have multiple replication origins (Gao, 2015; Ohbayashi et al., 2016). In addition to growth estimation based on coverage depth, it is also difficult to predict replication origins from sequence features such as GC-skew in some microbes (Gao & Zhang, 2008; Sernova & Gelfand, 2008; Vieira-Silva & Rocha, 2010). To overcome this challenge, a previous report proposed a method for predicting the positions of multiple replication origins based on the amount of chromosomal DNA (Xu et al., 2012). However, no method with a statistical background has been introduced. Third, the characteristics of coverage depth distributions themselves have not been sufficiently investigated. Some previous studies have reported non-linear DNA quantity trends (Hawkins et al., 2013; Pelve et al., 2013; Wu et al., 2014; Akiyama et al., 2016), such as those including a significant increase around the replication origin. Based on these studies, it has been suggested that replication affects not only the ratio of maximum to minimum depth, but also changes the degree of density around the replication origin. Modeling this phenomenon could be useful for both molecular biologists and microbiologists, enabling them to quantify the extra dynamics of replication. Furthermore, it is unclear whether the coverage depth trend is skewed toward the 5′direction, is skewed toward the 3′direction, or is symmetric. As some previous studies have suggested that the asymmetry of replisome progression is associated with the phenotype (Rodriguez-Lopez et al., 2002), it would be valuable to develop a method of symmetric level detection.

Here, we propose a method of modeling coverage depth dynamics using probabilistic statistics. Focusing on data generation when mapping fragmented reads to a circular genome sequence, we combined multinomial and directional distributions to mimic the read sampling process and bias of the DNA quantity. When applied to a dataset from a culture experiment, our method provided a stable and robust estimation of even a small number of reads and mutated reference sequences. To observe the degree of correlation between the growth estimates and experimental growth rates, we applied our method to WGS reads, which were obtained from a previous time-series culture experiment (Korem et al., 2015); this led to the observation of a high degree of correlation between the growth estimates and experimental growth rates. In vivo data sets were used to confirm the reproducibility of the growth dynamics in previous studies. Using the previous in vitro and in vivo samples, we ensured that our method is sufficiently robust to coverage and noise. Furthermore, by extending these models to enable them to form tapered and skewed coverage depth shapes, we designed a method of measuring coverage depth bias. Using a mixture of directional distributions allows growth estimation to be applied to sequences with multiple replication origins. We also demonstrate such estimations in relation to genome sequences of Sulfolobus solfataricus and Haloferax volcanii (McCarthy et al., 2015).

Materials & Methods

Circular distributions and statistics

The distributions and statistics used in this study are shown in Table 1. The location parameter has the highest probability and corresponds to the replication origin in the chromosome in this model. We changed the character of the concentration parameter by changing the distribution, as it can be aligned by ρc.= tanhκ2 or ρw.C.= tanhκ2 (Jones & Pewsey, 2005; Pewsey, Neuhäuser & Ruxton, 2013). In addition to major distributions, we introduced a linear cardioid distribution and exponential linear cardioid distribution to evaluate the coverage depth trend. These functions are symmetric around the location parameter, and the integral around a unit circle is 1; i.e., ∫−ππPθ|μ,ρdθ= ∫02πPθ|μ,ρdθ=1. For each distribution, the probabilistic PTR can be analytically defined as the ratio between the maximum and minimum value of the probability density function (see the Statistical model to estimate replication rate section for details).

Table 1 Circular distributions for modeling coverage depth dynamics.

Name	(Probability density) function	pPTR	Parameters	Ref.	
von Mises (vM)	expκ cosθ−μ2πI0κ	exp2κ	μ, κ		
Cardioid (c)	1+2ρc cosθ−μ2π	1+2ρc1−2ρc	μ, ρc		
Wrapped Cauchy (wC)	1−ρwC22π1+ρwC2−2ρwC cosθ−μ	1+ρwC21−ρwC2	μ, ρwC		
Jones-Pewsey (JP)	coshκψ+sinhκψ cosθ1ψ2πP1ψcoshκψ	exp2κ	μ, κ, ψ	Jones & Pewsey (2005)	
Linear cardioid (lc)	1+2ρlcθ−μ−π−π22π	1+πρlc1−πρlc	μ, ρlc		
Exponential linear cardioid (elc)	ρelc exp2ρelcθ−μ−π−π2 expπρelc− exp−πρelc	exp2πρelc	μ, ρelc		
Mean resultant length (mrl)	∑t=1T cosθt2+∑t=1T sinθt2T				
Notes.

μ location parameter

κ or ρ concentration parameter

ψ shape parameter

I0 modified Bessel function of the first kind of order 0

P1ψ the associated Legendre function of the first kind of degree 1ψ

θ an angle converted from the observed position

T total number of observations

Some of the models (von Mises, cardioid, wrapped Cauchy, and Jones-Pewsey) were symmetrically or asymmetrically extended with or without inverse transformation, as previously described (Abe, Pewsey & Shimizu, 2013; Pewsey, Neuhäuser & Ruxton, 2013; Abe, Pewsey & Fujisawa, 2013). To make the shape near the mode of a distribution variable, we used Batschelet or inverse-Batschelet transformation. Batschelet transformation (symmetric extension; SE) transforms the angular variable into gλθ=θ−μ+λsinθ−μ, where λ is the peakedness parameter. Using this transformation, the normalization constant was calculated using the composite Simpson’s law as the integral around a unit circle cannot be maintained as 1. Inverse-Batschelet transformation (inverse symmetric extension; InvSE) transforms the angular variable into gλθ=1−λ1+λθ+2λ1+λt1,λ−1θ, where t1,λθ=θ−121+λsinθ−μ. This transformation does not change the normalization constant. To make the distribution asymmetric with mode-invariance, we used the mode-invariance asymmetric transformation extension (MIAE) or inverse-transformed mode-invariance asymmetric transformation extension (InvMIAE). These transformations satisfy the requirement that asymmetricity be analyzed in replication. As replication begins at the origin irrespective of the rapidity of bacterial growth, the highest coverage depth position is preserved regardless of the asymmetry level. The skewness parameter must not affect the pPTR when the pPTR and skewness are measured independently. In these transformations, the symmetricity around the mode changes via an additional skewness parameter, while the location parameter, concentration parameter, and pPTR remain unchanged even if the skewness parameter changes. MIAE transforms the angular variable into gνθ=θ−νsin2θ−μ, where ν is the skewness parameter. This transformation also requires a normalization constant. InvMIAE transforms the angular variable into gνθ=sν−1θ, where sνθ=θ+νsin2θ−μ. This transformation does not change the normalization constant, and the position of the mode is preserved. To compute the inverse transformation, we used several root-finding algorithms (see Parameter estimation section). The significance of fitness improvement via additional parameters was evaluated using the likelihood ratio test with a chi-squared distribution based on the theorem of Wilks in addition to the Akaike Information Criterion (AIC) and the Bayes Information Criterion (BIC). To compute the likelihood ratios, the original distribution was compared with the extended distributions. The Jones-Pewsey distribution was compared with the von Mises distribution.

Statistical model to estimate replication rate

A statistical model that simulates the coverage depth dynamics along the genome position was constructed. Let di,s be the coverage depth of the i th position obtained when mapping the WGS reads of sample s onto the genome sequence. Here i represents the binned position of coverage. If the coverage depth is not compressed or binned, i matches a nucleotide position. We fit a generalized linear model (GLM) for each di,s as follows.

The starting point of our model is the conversion of di,s into the frequency of observation of the i th position. Here, the total number of observations Ts for sample s is calculated as the sum of di,s over sequence length, I, i.e., Ts=∑i=1Idi,s. We did not directly fit the coverage depth to a model because it could fail to fit with a low coverage depth dataset. Instead, we modeled a bias to observe the base positions by mapping reads with a probability distribution P and parameter set ω, which defines the potential of observation probability, it,s∼Pωs, where t is a unique identifier of nucleotides for all reads mapped to the genome. Focusing on the genome structure of bacteria and archaea, the observation probability is supposed to be circular. For compatibility with the structure, the position i is converted into an angle θ, following θ=iI2π. Here, the coverage depths di are stacking counts of the observed angle θ. Circular statistics, instead of an ordinal real-value approach, are required to quantify the bias based θ. In circular statistics, the first possible means of analyzing an angle dataset involves expressing the bias as a simple statistic without any model. For example, the mean resultant length (MRL) represents how data are concentrated around the sample mean direction. The second approach, which was mainly used in this study, involves the modeling of a phenomenon via probability distributions that generate positions. We introduced the following four distribution types from the circular distribution family: von Mises, wrapped Cauchy, cardioid, and Jones-Pewsey distributions. These distributions are widely used in circular statistics and are versatile in terms of implementation and inference (Jones & Pewsey, 2005; Pewsey, Neuhäuser & Ruxton, 2013). Additionally, these distributions are useful for representing known coverage depth characteristics. For example, some researchers have described non-linear coverage depth trends over genome sequences in both bacteria and archaea (Chen et al., 2005; Watanabe et al., 2012; Hawkins et al., 2013; Pelve et al., 2013; Rudolph et al., 2013; Wendel, Courcelle & Courcelle, 2014; Wu et al., 2014; Maduike et al., 2014; Yang et al., 2015; Akiyama et al., 2016; Ohbayashi et al., 2016; Forsyth et al., 2018; Retkute et al., 2018). It was expected that this trend could be quantified by extension for the distributions proposed in the previous studies. For these circular distributions with likelihood Ldist., the overall log-likelihood logL of the model can be calculated as follows: (1) logLω|θ,d,I,S= ∑s=1S ∑i=1Idi,s logLdist.ω|θi.

As θ is considered to be continuous rather than discrete for the purpose of these distributions, we confirmed that the parameters could be estimated appropriately (Text S1). Equation (1) coincides with a term that changes with the probability parameter in the log-likelihood equation of the multinomial distribution shown in Eq. (2) although the sum of the probability over all nucleotides is not 1 because the distribution is not discrete: (2) logMultinomiald|T,p= ∑s=1S ∑t=1Ts logt−∑s=1S ∑i=1I ∑j=1ds,i logj+∑s=1S ∑i=1Ids,i logpi.

Following the model with the likelihood represented by Eq. (1), the location parameter corresponds to the position of the replication origin as long as the concerned chromosome does not have multiple replication origins. Contrastingly, the concentration parameter is associated with growth as it determines the shape of the distribution. Therefore, we allowed the location parameter to be shared among all of the samples and the concentration parameter for each sample to be independent. For the concentration parameters, we set the half-Student’s t-distribution as a prior distribution (Gelman, 2006). We set 2.5 as the shape parameter; 0 as the location parameter; and 0.2 (von Mises, Jones-Pewsey), 0.1 (cardioid), 0.17 (wrapped Cauchy), 0.105 (linear cardioid), and 0.1103(exponential linear cardioid) as the scale parameters. These were selected such that the value of the cumulative probability density function became nearly 0.8 when the PTR was 2.0. This characteristic suggests that most of the PTRs are distributed between 1.0 and 2.0 in an environment. The distribution of the coverage depth PTR in a previous study rationalizes this suggestion (Korem et al., 2015). For the degree of density of the Jones-Pewsey distribution, the peakedness of the symmetric extended distribution, and the skewness parameter of the asymmetry extended distribution, we set the Gaussian distribution with a location parameter of 0 and a scale parameter of 1.0 as the prior distribution to avoid overfitting. From the model, we introduced an estimate that expresses the degree of growth. It is known that many microbes in prokaryotes replicate their chromosomal DNA on both sides from the origin such that the apparent amount of DNA increases near the origin. This behavior introduces a latent bias that makes DNA near the origin more likely to be observed during replication. This bias is simply expressed by the concentration parameter in a circular distribution. However, for consistency with the previous study (Korem et al., 2015), we defined a probabilistic PTR (pPTR), which is the ratio of the maximum of probability density function to the minimum, i.e., PTRprobability=pmaxθpminθ, as a growth dynamics index. This score represents the latent bias of the probability for the position at which a nucleotide is observed around the replication origin. Unlike the original PTR, which is directly estimated from the coverage depth, pPTR is obtained by modeling the bias based on a circular distribution and the probability framework of interest. Following the model, in which the coverage depth is a result of discrete sampling expressed as in Eq. (2), the coverage depth at a given position can be modeled by the binomial distribution by formula (3): (3) di,s∼BinomialTs,Pθ=iI2π|ωs.

This equation hints at the benefit of using probability rather than coverage depth directly (Text S2; Fig. S1).

Extending the model to multiple origins of replication

We constructed a statistical model for multiple origins of the replication-based mixture model and circular distributions. Let αm be the ratio of the mixture to the m th replication origin and M be the number of the replication origin, then the probability of obtaining the angle θ from the sample s is formulated to Pθ|ωs,α=∑m=1MαmPcircleθ|ωs,m with the circular distribution Pcircle. Note that the sum of the ratio is 1, i.e., ∑m=1Mαm=1. Based on the model, the overall log-likelihood logL can be calculated as follows: (4) logLω,a|θ,d,I,S∝∑s=1S ∑i=1Idi,s log∑m=1MαmLdist.ωm,s|θ=iI2π,s=s.

The equation is compatible with a genome sequence with a single replication origin as it takes the same form as Eq. (1) when we set α1 = 1 and M = 1. We set a Dirichlet distribution as the prior distribution of α as α ∼ Dirichlet(A) and employed 50∕M as A, as previous studies have implied that each replication origin shows similar activity (Robinson et al., 2004; Andersson et al., 2010; Hawkins et al., 2013). Then, the ratio is likely to assume a similar value, which defines the equality of the mixture. As the activity index for multiple replication origins, we defined a weighted pPTR (wPTR) and mean-weighted pPTR (mwPTR). The wPTR of the m th replication origin is computed via a weighted concentration parameter using a mixture ratio, where mwPTR is the average of these. For example, the wPTR of the von Mises-based model is given by exp(2αmκm), and mwPTR is given by 1M∑m=1MwPTRm. These scores are based on the model that replication stops if the replisome comes across another replisome, as mainly reported in prokaryotes (Leman & Noguchi, 2013; Wendel, Courcelle & Courcelle, 2014); however, this model has not been sufficiently investigated in archaea or eukaryotes. This model assumes that the effect of multiple origins at each location can be expressed as their sum. Following the mixed effects of multiple origins, the coverage depth is probabilistically sampled. This assumption results in the probability of regions that are not related to the origin approaching low values. Hence, the probability distribution of each origin becomes very steep, with increases in the concentration parameter and unweighted PTR. By weighting the parameters, we approximated the degree of activity in the case in which only the single origin worked in the chromosome. We used the average of the wPTR as the representative score of the chromosome because a previous study reported that the growth rate decreased when a replication origin on the chromosome was knocked out, whereas the deviation of the DNA amount between the origins and terminus did not change significantly (Wu et al., 2014). By using the average, the effect of activation on multiple origins could be determined.

Parameter estimation

For all data, we estimated the model parameters using an implemented software package. This package fits the parameters to the data by maximizing the joint posterior (optimizing mode) or generating samples from the posterior distribution of the parameters (sampling mode). Briefly, after maximizing the log-likelihood of the model for the data via each method, we adopted the value that yielded the maximum log-likelihood via the optimizing mode or the expected a posteriori (EAP) of the parameter posterior distribution via the sampling mode. Unless otherwise noted, we used the limited-memory Broyden–Fletcher–Goldfarb–Shanno (L-BFGS) algorithm for the optimizing mode and the No-U-Turn Sampler (NUTS) algorithm (Hoffman & Gelman, 2014), which is the quasi-Markov chain Monte Carlo (MCMC) method, for the sampling mode. For sampling, we set the number of chains to 1 and the number of iterations to 500; the first 300 iterations were considered a warm-up and discarded. From the samples, we calculated the EAP as a representative model parameter. The convergence of the sampling was checked using Rhat statistics. If the statistics were equal to or less than 1.1, the result was accepted (Gelman et al., 2013). We used composite Simpson’s algorithm with 20 subintervals to calculate the integrals in the models. We computed the inverse transformation of the functions using Newton’s method when the skewness ν or peakedness parameter λ was less than or greater than 0.8. These thresholds were evaluated by performing manual simulations such that the transformations did not oscillate. We checked the convergence of the function as to whether the error was less than the machine precision of a float-type variable defined using Stan. The Illinois method was used in other cases (Dowell & Jarratt, 1971). Then, we checked the convergence of the function, as to whether the error was less than 1. 0 ×10−13. In both methods, we defined a maximum iteration count that terminated the calculation at 30 iterations for Newton’s method and 100 iterations for the Illinois method. We set the mathematics transformation to reasonably estimate the parameters. If a location parameter is estimated directly from 0 to 2π, and the true location parameter is located near the edge of the range, the parameter estimation is likely to fail as it does not detect cyclicity. We re-parameterized the location parameter as μ= arctan2θμ → for the two-dimensional unit vector θμ → to overcome this estimation difficulty (Pewsey, Neuhäuser & Ruxton, 2013). The unit vector can be estimated directly as each element has continuity from -1 to 1. To calculate information criteria, 1+number of the sample ×2 was used as the number of parameters for the Jones-Pewsey distribution-based model, and 1+number of the samples was used for the others.

Coverage depth calculation

The coverage depth was calculated by aligning the WGS reads to the template genome sequence. We downloaded WGS via the SRA Toolkit. After converting the WGS reads into the FASTQ format, we aligned them to the genome sequence using Bowtie2 with a “–very-sensitive” parameter set (Langmead & Salzberg, 2012). We sorted the resulting SAM files and calculated the depth using SAMtools (Li et al., 2009). Next, a moving median filter with a 100 nt window size and a 100 nt stride length was applied. The moving median filter runs through the coverage depths, replacing each coverage with the median of neighboring locations. We applied the moving median filter for the following two reasons: to reduce the noise and outliers and to reduce the data size, which affects the computational time of the model fitting. If the coverage depth seemed to have noise regions that increased the coverage due to highly conserved regions such as ribosomal genes, an additional filtering for outliers was performed; specifically, the top 1% of the coverage depth was removed and replaced with blank coverage. This decision and the threshold, which were determined based on a previous study (Brown et al., 2016), were independently evaluated using a frequency histogram of the coverage depth containing a noise region, which increases the coverage in multiple datasets (Text S3). Certain regions that remained blank following filtering were filled with 0. As the WGS reads of H. volcanii were separated into multiple FASTQ files, we concatenated them into a single file based on growth conditions prior to alignment. To evaluate the error in the sequence edge parts, we copied 263 nt in the head portion of the genome sequence to the tail portion. We also constructed a graph genome sequence, circularized it, and mapped WGS reads to the graph genome sequence using the variation graph toolkit (vg) with the default parameter set (Garrison et al., 2018). We omitted the GC-content correction procedure to simplify the pipeline as the method using the full-length genome sequence seemed to be robust (Brown et al., 2016) against the local coverage depth bias attributed to the GC-content (Ross et al., 2013). If a user requires greater accuracy, we recommend using the correction method or PCR-free library preparation (Benjamini & Speed, 2012; Brown et al., 2016; Gao & Li, 2018).

Growth rate evaluation with experimental growth rate

The accuracy of the growth estimates was evaluated via comparisons with experimental growth rates. Unless otherwise noted, the experimental growth rates were calculated from the colony formation unit (CFU)/ml, optical density (OD), or relative abundance using gri= log2abuni+1− log2abuni−1ti+1−ti−1 following the approach taken in a previous study (Korem et al., 2015); e.g., the experimental growth for Fig. S3C uses this equation. For relative abundance, we used the mOTUs2 pipeline with the default parameter set (Milanese et al., 2019). When comparing methods, we additionally calculated the experimental growth rate (shown in Fig. S3D) in a differential manner to obtain the dynamics in a short time span; i.e., gri= log2abuni+1− log2abuniti+1−ti. For comparison, we performed growth estimation using tools developed in previous studies (Korem et al., 2015; Brown et al., 2016; Emiola & Oh, 2018; Gao & Li, 2018). We added a time delay for the correlation coefficient between the experimental and growth estimates following the approach of the previous study (Korem et al., 2015). The time delay, which provided three or more combinations of the growth estimates and experimental growth rates and yielded the highest correlation coefficient, was accepted.

Effect of normalization on the model

We evaluated the effect of normalization on the parameter estimation by checking the difference between an estimated distribution and the true one. For the evaluation datasets, we generated continuous angles from the von Mises distribution. These angles were binned into discrete angles following the defined discrete length. We selected the location parameter from −π, −π2, 0, and π2; the concentration parameter from 0.1, 0.4, and 0.7; the discrete length to be segmented from 5, 10, 30, 120, 600, and 1,000; and the average coverage depth to be observed from 0. 5 ×, 1. 0 ×, 2. 0 ×, 4. 0 ×, 8. 0 ×, and 16. 0 ×. Next, we fitted the unnormalized and normalized von Mises distribution-based models to the simulated data. Finally, we evaluated the error of the estimated distribution using the Kullback–Leibler divergence (Kullback & Leibler, 1951). When evaluating the parameter of interest, the other parameters were fixed to 0 for the location parameter, 0.7 for the concentration parameter, 30 for the discrete length, and 16 for the average coverage depth. For the distribution with the true parameter, we set 10,000 as the discrete length. Calculations were performed 10 times with different seeds for each parameter set. This part of the procedure was performed using Scipy (Virtanen et al., 2019). We introduced a constraint cω to normalize the continuous circular distribution, i.e., Pdiscreteθn|ω=1cωPcontinuousθ|ω for discrete circular data θn(n = 1, 2, …, N) and the parameter set ω. For the von Mises and wrapped Cauchy distributions, we calculated the sum of the likelihood directly as logcω= log∑n=1NPθn|ω because we could not have a closed-form equation. For the cardioid distribution, it was formulated as cω=∑n=1N12π1+2ρcosθn−μ=N2π owing to ∑n=1N cosθ=0. For the linear cardioid distribution, it was formulated as cω=∑n=1N12π1+2ρθ−μ−π−π2=N+22π. The rearrangement of the formulas was performed using Sympy (Meurer et al., 2016).

Simulation of skewness after coverage depth sorting and investigation of the causes

The cause of the skewed shapes appearing at both ends of the coverage depth after sorting required investigation. The probable phage and duplicate gene regions are considered to generate outliers and to form the skewed shape. Thus, the probable regions on the chromosomal sequence were annotated. To identify probable phage regions in the genome, we used PHAST (Zhou et al., 2011). For the duplicate gene regions, first, we used Prokka to predict the coding sequences (Seemann, 2014). Next, we mapped these predicted sequences to the genome sequence of Lactobacillus gasseri using Bowtie2 and annotated regions as overlapping if two or more hits were obtained. The parameter set of Bowtie2 was “-a –very-sensitive.” Thereafter, we extracted the partial coverage depth from 1.0–1.4 Mnt regions in which annotated features did not exist. To obtain the odds ratio, 36 metagenomic sequences obtained from the previous study (Korem et al., 2015) were mapped, and the coverage depth was calculated via the method described above. Next, the coverage depth for each sample was sorted, and the number of annotated bases in 5% of the upper, lower, and total sequence was counted. We modeled the number n to follow a binomial distribution with the total number of bases N of the target sequence and the appearance probability p as parameters: n ∼ Binomial(N, p). The odds ratio was calculated as odds=p1−p from the appearance probability p. To estimate these parameters, we employed an MCMC algorithm with NUTS using PyStan. Four chains were utilized, and 20,000 iterations were performed, where the first 1,000 of these were discarded as warm-ups. The posterior distribution of the EAP was used as the representative.

Evaluation of robustness in terms of coverage depth using culture dataset

To investigate the robustness of our proposed method, we compared the growth estimates calculated using a sufficient amount of reads and those using rarefied reads. To evaluate coverage depth dynamics from a single origin, we first used L. gasseri WGS samples with an average of more than 20 × coverage (n = 20). After confirming that many variations occur at lower than 5. 0 × coverage, we selected Escherichia coli, Enterococcus faecalis, or L. gasseri WGSs reads with more than 5 × coverage from the dataset by Korem and colleagues. To evaluate multiple origins of replication data, the WGS of S. solfataricus was used. We randomly sampled reads from the FASTQ files using seqtk (Li, 2013) such that the average coverage depth would be 0. 001 ×, 0. 005 ×, 0. 01 ×, 0. 05 ×, 0. 1 ×, 0. 5 ×, 1 ×, or 5 ×. In the first evaluation of a replication dataset from a single origin, we additionally sampled 10 × and 20 × coverage. In the evaluation of multiple origins, we additionally sampled 100 × coverage. The pPTR, wPTR, and mwPTR were calculated using the rarefied reads and compared with those obtained from 20 × and the full coverage depth. For the S. solfataricus dataset, we specified the number of components as three and performed estimation via the optimizing mode using 30 different seeds. We selected representative results with the highest likelihood and compared pPTR and mwPTR with those corresponding to no modifications. As DEMIC cannot work with a single genome sequence even if it is complete, we used a genome sequence obtained by co-assembling all of the reads using MEGAHIT with the default parameter set (Li et al., 2015). We utilized the default parameter set for PTRC, DEMIC, bPTR, iRep, and GRiD. Finally, we calculated the error rates as (5) Estimatemodified−EstimatereferenceEstimatereference

The results from original WGS reads were used for reference estimates. To validate the error rate, we defined 15% as a threshold, as was done in a previous study (Brown et al., 2016).

Evaluation of robustness in terms of coverage depth using metagenomics reads

To evaluate the robustness using metagenomic datasets, we employed the inflammatory bowel disease (IBD) dataset from a previous study (Franzosa et al., 2018). We searched for combinations of species and WGSs with an average cover depth of 20 × or more. In order for the first screening to satisfy the scope of the method, we used Kraken2 (Wood, Lu & Langmead, 2019) and Bracken (Lu et al., 2017) with the default parameters; the objective of this was to count the number of reads to be assigned to the genome sequences. As a collection of complete chromosomal sequences, we constructed a database with species-level resolution (see Complete genome sequence database section). Based on the taxonomic profile, the combinations with more than 0. 1 × coverage depth were selected as candidates (n = 16,413 combinations from 220 WGS samples) in the first screening. Next, we aligned the WGS reads of the datasets to the database using Bowtie2 and calculated the coverage depth using SAMtools. After applying the moving median filter and outlier elimination, we calculated the average coverage depth. If the combination had more than 20. 0 × average coverage and passed the first screening, we concluded that the combination was eligible to serve as an evaluation target (n = 676). For the targets, we extracted paired-end aligned reads using SAMtools with the “view -f 2″option for PTRC. This procedure was required because the GEM-mapper, which is used in PTRC, did not allow singletons to be aligned. After that, we rarefied the reads using SAMtools with the “view -s” option and converted the alignment result into a FASTQ file for the input. For our method, we counted the coverage depth from the alignment result using SAMtools. After applying each method, we calculated the error rate following (5). We used PTRC as a benchmark method, as it provided the most stable estimation with low coverage in the culture WGS dataset.

Evaluation of robustness in terms of mutation rate

The robustness of our method in terms of the mutation rate with a single replication origin was assessed by evaluating how much of the value was maintained when the reference genome was used for mutation estimation. The reference genome sequence was mutated in three ways. The first involved nucleotide-level mutation at random positions in the genome sequence. Based on the length of the genome sequence, every 5% portion, ranging from 5 to 30% of the nucleotides, was randomly selected and mutated to an ambiguous nucleotide N. The second way involved block-level mutation at random positions in the genome sequence. We used msbar in EMBOSS to create block-level mutated sequences (Rice, Longden & Bleasby, 2000). Based on the length of the genome sequence, every 5% portion, ranging from 5 to 50% of the nucleotides, was randomly mutated. We used 5,000 nt as the block size. The third way involved mutation at the block level at a specific region in the genome sequence. We randomly selected the position to be mutated. The size of the region was determined based on the sequence length. After that, every 5% portion, ranging from 5 to 30% of the nucleotides, was mutated to an ambiguous nucleotide N. As the positional relationship of the contig sequence was unknown, the evaluation of DEMIC in this regard was not performed. The first and third mutations were performed using in-house scripts with Biopython. After constructing pseudo-mutated genome sequences, we estimated and compared the growth following the above procedure. We assessed the robustness of the estimation with multiple replication origins, mutating the S. solfataricus genome sequence using the first and second methods. The estimations and evaluations for multiple origins of replication were performed in the same manner as the robustness evaluation with low coverage depth.

Evaluation of robustness in terms of peak noise

To investigate the influence of peak noise, we generated an artificial dataset. We used the short-read sequence of E. faecalis, L. gasseri, and S. solfataricus published previously (Korem et al., 2015; Payne et al., 2018). Firstly, a 100 bp region was randomly selected from a reference genome sequence using the seqkit sample command. The sequence was copied every 10 times from 10 to 100 times and added to the FASTQ files. We set 93 as the quality score. Using the mixed file, we calculated the coverage depth and estimated the growth in optimization mode following the procedure described above. For filtering, the top 1% of the depth was removed. Following this step, we computed the error rate using Eq. (5) and compared the results with the noiseless results. The evaluation for multiple replication origins was performed in the same manner as the evaluation for low coverage depth. Secondly, we randomly selected 1,000 bp regions from the reference genome sequence every 10 regions from 10 to 100 regions. This length was determined based on the average gene length of prokaryotes. These sequences were mixed with the WGS such that the coverage depth amounted to 20 ×, 40 ×, 60 ×, 80 ×, and 100 × in each region. Finally, we evaluated the error of the estimates following the same procedure.

Evaluation of robustness in terms of sample size

To assess the effect of the sample size on the estimates, partial sample sets were generated from the full sample set, and the results were compared. We used the short-read sequences of E. coli, E. faecalis, and L. gasseri published previously (Korem et al., 2015). The partial set was configured to include 1, 4, 8, 12, 16, and 20 samples. Each set was distributed in a manner that avoided duplication of the same sample. Except for the sample set with only one sample, the sets were constructed to contain each sample at least 10 times. For each sample set, preprocessing and inference were conducted according to the above procedure, and the error rate was calculated in comparison with the results obtained when all of the samples were used simultaneously.

Skewness in Watson and Crick strands

To count the coverage depth in Watson strands and Crick strands, we used the SAMtools view command with the “-f” option set to 0 for the Watson strand and 16 for the Crick strand. This was done after mapping reads to the template genome sequence. The procedures that followed were the same for both strands. Finally, the highest log-likelihood in 30 independent trials was used as a representative estimate.

Growth estimation of species with multiple replication origins

We used the von Mises distribution for mixing because it has an intermediate degree of density around the mode and an open range of concentration parameters; i.e., κ > 0. Both genomic and short-read sequences were obtained according to the procedures described previously (Ausiannikava et al., 2018; Payne et al., 2018). The coverage depth was calculated according to the above-mentioned procedure. As the deletion was confirmed in the genome sequence of S. solfataricus by the Integrative Genomics Viewer (Robinson et al., 2011), we deleted regions from 1,443,200 nt to 1,485,069 nt on CP011055, from 1,443,192 nt to 1,485,075 nt on CP011056, and from 1,443,197 nt to 1,485,072 nt on CP011057. We used Dfast to annotate cdc6 in the genome sequences (Tanizawa, Fujisawa & Nakamura, 2018). The number of components M was determined by using AIC and Widely Applicable Information Criterion (WAIC) as the mixture model was a singular model, and there was a possibility that a decision based only on AIC could produce incorrect results. The MCMC algorithm was applied to the constructed model, and WAIC was calculated from the log-likelihood (Watanabe, 2010; Gelman et al., 2013). The calculation was performed using an in-house Stan script and CmdStan with the threading option. The sampling was performed 1,500 times on a single chain, where the first 1,000 samplings were excluded as warm-ups. We fitted the model to the data assuming a number from 1 to 4 for the distribution. To calculate the AIC, we used the EAP of the posterior distribution to represent the log-likelihood.

Growth rate estimation for infected Citrobacter rodentium

x WGS reads of mice fecal samples from the original growth dynamics analysis study were used (Korem et al., 2015). These sequences were aligned to a complete genome sequence database (see Complete genome sequence database section) to validate the applicability of using multiple reference genome sequences using Bowtie2 with the “–very-sensitive” option. After that, we counted the coverage depth by SAMtools. After extracting the coverage of the C. rodentium chromosome sequence, we cleaned and compressed the coverage using the moving median filter after removing the top 1%. Finally, we fitted the von Mises model to the coverage. The pPTRs were compared via Welch’s t-test.

Complete genome sequence database

The Genome Taxonomy Database version 89.0 was used to control the fineness of the taxonomy on the species level (Parks et al., 2018). For each species, when the representative species had a complete genome, it was used. When it did not, the sequence with the highest CheckM completeness score (Parks et al., 2015) and largest genome size was used. Species without complete genome sequences or those with multiple chromosome sequences were excluded. Mobile genetic elements were excluded from the database by checking the sequence label using seqkit; we filtered out the sequences labeled “plasmid,” “Plasmid,” “phage,” “chromid,” “pMLa,” and “Linear.”

Growth estimate evaluation on metagenomic dataset

The growth dynamics were estimated at species-level resolution. We filtered low-quality reads in WGS via Trimmomatic and then removed human-derived reads by aligning them to the GRCh38 reference human genomic sequence using Bowtie2. We used “SLIDINGWINDOW:4:15 MIN LEN:36” as a parameter in Trimmomatic. After quality control, we aligned qualified metagenomic reads with the complete genome sequence database using Bowtie2 with the “–very-fast” option. After extracting the alignment results of the target reference sequence, we counted the coverage depth. As the metagenomic sequences were not clean compared to the culture datasets, we performed additional filtering as described in the coverage depth calculation method. After preprocessing was completed, we fitted the model to the coverage depth of each sequence via the optimizing mode. For the estimations, we selected samples with greater than or equal to 0.0001% relative abundance of the taxon and greater than 0. 01 × average coverage depth. We used Kraken2 and Bracken with the complete genome sequence database to estimate the relative abundance of the species. After filtering out the ultra-low coverage depth samples, we excluded the samples that might not achieve random sampling from the chromosomal DNA sequence. This is because the estimates of these samples would have an error. To detect the invalid samples, we focused on the difference of actual zero coverage fraction f and a theoretical score f ˆ based on the Lander-Waterman theory (Text S4). This theoretical score can be obtained as f ˆ≈exp−a, where a denotes the average coverage depth. For samples with average coverage less than 5. 0 ×, we excluded samples with log-scale fractions greater than 0.56 times the theoretical score. Assuming there to be uneliminated noise coverage depth, samples with estimated PTRs greater than or equal to 3.0 were excluded. Welch’s t-test for independent groups was used to examine the differences between the growth estimates, and Hedges’ g was used to evaluate the effect size for the two groups.

Software

We implemented the statistical model using Stan (Carpenter et al., 2017). Wrapped by Python scripts, this model is available for use in the command-line environment. This package also contains a moving median filter, a visualizer, a statistics profiler based on directional statistics, an information criterion calculator with estimated results, an asymmetric test calculator using Pewsey’s method, and other utilities required to analyze the coverage depth over replicon. Other software versions are summarized in Table S1. Our package for growth estimation is available from https://github.com/TaskeHAMANO/SPHERE. This software was implemented using Python3 (≥3.6) and Stan. The wrapper software used in this study for PTRC, DEMIC, and GRiD is available from https://github.com/TaskeHAMANO/PTRC-in-cwl, https://github.com/TaskeHAMANO/DEMIC-in-cwl, and https://github.com/TaskeHAMANO/GRiD-in-cwl, respectively. This software is distributed under the BSD-3-Clause license. The wrapped software of msbar in EMBOSS is available from https://github.com/TaskeHAMANO/msbar-in-cwl This software is distributed under the GPL-3.0 license. These wrapper scripts were implemented using the Common Workflow Language (CWL) v1.1. These scripts have been tested on Linux and macOS.

Availability of data and material

The WGS data of time-series-cultured E. coli, E. faecalis, L. gasseri, S. solfataricus, and H. volcanii are available from BioProject (PRJEB9718, PRJNA250819, PRJNA250820, PRJNA250827, PRJNA346830, PRJNA250832, PRJNA250833, and PRJNA422812). The genome sequences of E. coli NMC3722, E. faecalis ATCC 29212, L. gasseri ATCC33323, S. solfataricus SULA, SARC-B, SARC-C, USLG, SARC-H, SARC-I, and H. volcanii DS2 are available from GenBank and RefSeq (CP011495, CP008816, NC_008530, CP011057, CP011055, CP011056, CP033235, CP033236, CP033237, and NC_013967). The genome sequence of H. volcanii H26 was modified from DS2 as previously described (Hawkins et al., 2013). The genome and metagenome sequences used in the cohort studies analysis are listed in Table S2. The final chromosome sequences we used to construct the genome sequence database are listed in Table S3.

Results

Creating an artificial coverage depth

We constructed a statistical model for coverage depth dynamics based on circular distributions. To validate our model visually, we generated an artificial coverage depth using the above-mentioned circular distributions (Text S5). The generated coverage depth reproduced high variance and concentration at the replication origin, expressed as the location parameter of the circular distribution. Interestingly, when sorted, this artificial coverage depth showed a distorted trend in both the upper and lower orders regardless of the circular distribution type (Fig. 1A). The same shape was visualized previously (Brown et al., 2016), wherein it was stated that this shape was formed by a specific sequence feature, such as a phage. However, our model generated artificial depths from smooth probability trends and did not include any artificial noise. To investigate in detail the cause of the distorted regions seen at both ends, potential prophage sequences and duplicate genes in the genome sequence of Lactobacillus gasseri were analyzed. Among them, only the intact prophage region was abundant at the lower end (Table S4). Moreover, a similar distorted structure was reproduced on a partial genome sequence that did not contain suspicious regions (Fig. 1B). We evaluated the best model for this distribution and used it to determine the threshold for outlier removal (Table S5; Text S3).

Figure 1 Effects of growth, sequence feature, and outliers on coverage depth shape.

We characterized the coverage depth of chromosomal DNA using statistical models. The shape of the probability distributions (solid lines) and artificial coverage depth (blue lines) obtained using the (A) von Mises, (B) cardioid, (C) wrapped Cauchy, (D) Jones-Pewsey, and (E) linear cardioid distribution model with multinomial distribution. Zero (0) was used as a location parameter, while 0.34657 (von Mises and Jones-Pewsey), 0.16666 (cardioid), 0.17157 (wrapped Cauchy), and 0.1061 (linear cardioid) were used as concentration parameters to align the pPTR with 2.0. The nucleotide number was set to 1 Mnt, and the average coverage depth was set to 20 × in the multinomial distribution. For the Jones-Pewsey distribution, 0.5 was used as the shape parameter. Sorted shapes of the distributions and pseudo-coverage depths from the (F) von Mises, (G) cardioid, (H) wrapped Cauchy, (I) Jones-Pewsey, and (J) linear cardioid distribution model with multinomial distribution. (B) Coverage depth and sequence features that can cause strong noise in the coverage depth of L. gasseri (ERR969426). (K) Overall coverage depth, (L) suspected feature-free region, (M) sorted overall coverage depth, and (N) sorted feature-free region.

Performance evaluation with experimental growth rates

Using a statistical model based on a circular distribution, we first evaluated the model’s accuracy by estimating the correlation between the computational and experimental growth rates, as had been done previously (Korem et al., 2015). We estimated the coverage depth by mapping the WGS reads to the genome sequence and counting the coverage depth (Fig. 2A). After reducing the variance, outliers, and data size with a moving median filter, the proposed model was fitted to the cleaned coverage depths (Fig. 2B). To evaluate the accuracy of the method, we used the WGSs of the three species (E. coli, E. faecalis, and L. gasseri) previously obtained from culture experiments under aerobic and anaerobic conditions (Korem et al., 2015). These data were accompanied by CFU/ml or OD in time series for evaluation. As substantiated in the previous study, we observed a high correlation coefficient between the growth estimates and experimental growth rates (Fig. 2C; Figs. S2A and S2B). Regardless of the culture state, E. coli and E. faecalis exhibited high degrees of correlation, without requiring time delay adjustments (r ≥ 0.5). In contrast, L. gasseri required a time delay adjustment of 90–120 min. Our growth estimates yielded correlation coefficients equivalent to those obtained using the previous methods, with experimental growth rates of both 60 min (Fig. S2C) and 30 min (Fig. S2D). Our estimates were correlated with the temporal growth based on the relative abundance (r = 0.76 ± 0.04, n = 4, each with 10 timepoints; Fig. S2E) previously obtained (Korem et al., 2015) even when the samples originated from mixed cultures with multiple intestinal species.

Figure 2 Probabilistic model for generating coverage depth.

Summary of procedures and distributions with coverage. (A) Overall flowchart of our method. The green parallelogram represents the data, and the pink rectangle represents the procedure. (B) Coverage depth on the circumference. Focusing on the genome structure of prokaryotes, we developed a model that conducted circular regression. The green and pink plots represent the peak and the trough estimated from the model, respectively. (C) Correlation with experimental growth rate by time series aerobic cultured E. coli. pPTR is correlated with the experimental growth rate with a Pearson correlation of 0.964. The dataset was obtained previously (Korem et al., 2015).

Secondly, we tuned the parameters of interest. The window and stride size of the moving median filter were optimized to 100 bp by comparing the growth estimates with the experimental growth rates (Figs. S3 and S4; Texts S5 and S6). Our pipeline, which used a sequence aligner that did not take circular structures into account, confirmed that the decrease in coverage at both edges, termed the edge effect in a previous study (Brown et al., 2016), exerted only a small effect on the estimation (Fig. S5; Text S7). Our method performed well regarding memory usage and computation time with the exception of the Jones-Pewsey distribution-based model (Fig. S6).

Finally, we evaluated the applicability of the method using artificial datasets. When applied to modified datasets from culture experiments, most methods, including those used in previous studies, have performed adequately with low coverage depth WGSs until at least 0. 5 ×, whereas our growth estimates remained stable even at coverage depths of 0. 01 × (Fig. 3A and 3S and Fig. S7 a). According to the bPTR in the coverage depth mode, the error percentage was not correlated with the number of reads. Although GRiD seemed to maintain a low error rate from 0. 005 × to 0. 01 × coverage depth, the number of samples with growth estimated as 1.0 increased (0. 005 ×: 10/20, 0. 01 ×: 17/20). As DEMIC requires multiple contigs for estimation and thus is not applicable to a single template genome sequence even when the sequence is complete, we performed the evaluation using assembled genome sequences. Moreover, this approach cannot be applied to low coverage less than or equal to 1.0 × coverage. Given the culture dataset results, we also performed the evaluation on metagenomic reads and confirmed that the error rate was less than 15% on average until 0. 01 × coverage (Fig. 3B). Our method was also stable for mutations at both the 5,000 bp block and single nucleotide levels (Figs. S7B and SC). Because the approach uses the entire sequence structure, the results obtained from a sequence mutated on a single specific region deviated from those of full-length sequences (Fig. S7D). To evaluate the effect in human intestine WGS reads, we quantified the deletion size on the chromosome sequences in a metagenomic dataset (Fig. S8; Text S8). Our estimates were as stable as those generated using the previous methods when a single peak noise was contaminated (Fig. S7E). With more noise, although the error rate of our estimates remained less than 15% on average, some samples showed substantial error as the amount of artificial noise increased (Fig. S7F). To address this, we investigated the relationship between the error rate and the zero coverage fraction (Lander & Waterman, 1988; Roach, 1995) and determined the threshold to exclude invalid samples with an error rate of more than 15%. As a result, we detected the noise-contaminating coverage samples with a recall score of 0.81 (Texts S4 and S9; Figs. 9 and S10; Table S6). Finally, the number of samples was related to the variation, but the effect was not substantial compared with those of the other factors (Fig. S7G).

Figure 3 Error rates of growth estimates from various coverage depths with respect to the full coverage depth.

The error rates for each average coverage depth were calculated with respect to the full coverage depth. (A) Only the E. coli, E. faecalis, and L. gasseri WGSs with greater than 5.0× coverage, or (B) species with more than 20× coverage in human fecal WGS datasets were used (Franzosa et al., 2018). The horizontal bar represents the 15% threshold of the error rate threshold. The black crosses on the bar represent the unavailability of the methods with respect to the coverage depth. The proposed models and statistics are shown in the black rectangle within the legend.

Performance evaluation using in vivo dataset

To evaluate the accuracy, we compared the growth estimates with the known growth dynamics using previous datasets. For the in vivo sample setting, we tuned the window size of the moving median filter based on the coefficient of variance and concluded that 100 nt was the best (Fig. S11). First, we checked the reproducibility of the growth estimates using C. rodentium-infected mice fecal samples. As was also reported previously (Korem et al., 2015), tir mutant C. rodentium had a higher pPTR than the wild-type (WT) strain (Fig. S12; p-value by Welch’s t-test between WT and mutant on days 6–9: 8. 72 ×10−5, nWT = 12, nΔtir = 12). Second, we evaluated the growth dynamics in the fecal microbiome in IBD patients (Franzosa et al., 2018). When we compared the estimates between Crohn’s disease subjects and healthy volunteers, we reproduced the significant high growth estimates of Eggerthella lenta in the patients (p-value: 1. 26 ×10−7, Hedges’ g =  − 1.21, nhealthy = 42, nCrohn = 49). Although this difference was limited to the remission and active patients in the small sample size dataset used in the previous study (Korem et al., 2015), we observed this difference in the large cohort dataset even by PTRC (p-value: 1. 31 ×10−2, Hedge’s g =  − 0.57, nhealthy = 35, nCrohn = 57). Finally, we confirmed the growth dynamics of Bifidobacterium breve and Bifidobacterium adolescentis in the neonates and their mothers fecal microbiome cohort (Bäckhed et al., 2015). It is well known that B. breve is abundant in infant guts, whereas B. adolescentis is abundant in adult guts. Moreover, a previous experimental study demonstrated that B. breve grows well in a medium containing formula based on soy, milk, or casein hydrolysate. These biological signals were also reproduced in the estimates obtained using our model (Fig. S13; Table S7).

Shape, peakedness, and skewness of coverage depth

As an additional application of our model, we investigated the shape of the coverage depth by comparing the kinds of circular distributions (Tables S8 and S9). In a comparison of the fitness of multiple models, the Jones-Pewsey distribution model exhibited the highest fitness among the vanilla models (those without argument transformation) on average. The shape parameter of the Jones-Pewsey distribution in the datasets of Korem et al. (2015) changes considerably with time (Fig. S14A). For example, in the E. faecalis dataset, the distribution was dense around the replication origin in the first phase; however, it gradually dispersed over time. In contrast, the trend was reversed in the anaerobically cultured L. gasseri.

To evaluate the coverage depth concentration phenomenon further, we implemented the InvSE von Mises distribution model. Along with the Jones-Pewsey distribution model, the peakedness parameter of the InvSE von Mises distribution changed considerably with time (Fig. S14B). Comparing the model with the vanilla von Mises-based model, the InvSE von Mises-based model exhibits a lower AIC and BIC (Table S8). As in the Jones-Pewsey model case, the peakedness was initially high. However, in E. faecalis, it became lower later on (Fig. 4). These parameters in the Jones-Pewsey and InvSE von Mises distributions showed high correlation coefficients, but their trends were not identical (min r =  − 0.793, p = 2. 93 ×10−7; Fig. 15).

Figure 4 Replication peakedness.

The InvSE von Mises distribution-based model exhibits a tapered shape of the coverage depth trend. The Jones-Pewsey distribution-based model also shows a concentrated shape.

Next, we evaluated the symmetricity of replication. Although several methods that extend circular probability densities toward asymmetricity have been described (Batschelet, 1981; Pewsey, 2002; Abe, Pewsey & Fujisawa, 2013), a few requirements must be satisfied to adapt to replication dynamics. Therefore, the InvMIAE von Mises distribution-based model was used in this study. First, we evaluated the robustness of the asymmetric extended method (Text S10; Fig. S16). As a result, we concluded that this extension was not suitable for use with the short-read sequences that were largely mutated from the template genome sequence. We therefore determined the applicability of the dataset by estimating the mutation rate from the frequency of the zero-coverage depth using a zero-inflated model (Supplementary Text S11). The results indicated that the E. faecalis dataset did not satisfy the criteria (Fig. S17A). Finally, we fitted the model to the actual coverage depth. The skewness parameter had a low variance with time and was nearly 0 except for the E. faecalis data (Fig. S14B). Although E. faecalis showed high skewness, the InvMIAE model fitted 0 or outlier coverage depths rather than the skewness of the whole sequence (Fig. S17B). Furthermore, we measured skewness using only Watson and Crick strands. The skewness parameters showed strong correlations, and no specific skewness in any specific strand was found (Fig. S18). These results support the hypothesis that coverage depth is symmetrical, contrary to our expectations.

Extension of the model to multiple replication origins

To demonstrate the extendibility of our method, we modeled the coverage depth behavior of multiple replication origins, using a mixture of circular distributions. To validate our model, we applied it to WGS data of Sulfolobus solfataricus and Haloferax volcanii, which contain three replication origins (Ausiannikava et al., 2018; Payne et al., 2018). Based on the AIC and WAIC, we determined the number of components in both datasets to be three because this number yielded the best score on average (Table S10). Of the seven datasets, five matched the true number of active replication origins. The estimated location parameters of S. solfataricus were distributed close to cdc6, which is a marker gene for the replication origin (the average error rate of the location parameter with respect to the marker gene is 6.55 ± 4.48%, n = 6) (Figs. S19A–S19F) (Lundgren et al., 2004; Robinson et al., 2004). In contrast, although there is a distinct peak around 2 Mbp, cdc6 is not evident in the H. volcanii genome sequence (Figs. 5A and 5B). We compared the weighted PTRs between the exponential growth and stationary phases. All of the origins in the exponential growth phase increased the estimates (exponential growth phase: 3.59, 3.18, and 2.66; stationary phase: 1.64, 2.26, and 1.43). We also checked the difference in the wPTR among multiple origins. Notably, the middle of the replication origin position nearly coincided with the position at which the genomes were split by the ratio of wPTR (Figs. 5C and 5D; Figs. S19G–S19L). As was done when the model was applied for a single replication origin, the robustness of the estimates was evaluated using the artificially modified dataset, which was an S. solfataricus dataset in this case. As a general trend, the individually weighted PTRs were more sensitive to the modifications than the mean weighted PTRs. When the number of reads was limited to 0. 1 × coverage depth on average, the error of the estimates was less than 15% at the median (Fig. 19SM). Although it was more susceptible to noise than the model for a single origin of the replication origin, this estimate was robust so long as the mutation rate was less than 7% at the point level or 4% at the block level (Figs. S19N and S19O). Our method avoided the effect of a single noise region which increases coverage in the conserved region (Fig. S19P).

Figure 5 Extension to multiple replication origins.

The plots in A and B represent the coverage depth and probability distributions estimated using a mixture of the von Mises distribution models for Haloferax volcanii in the (A) exponential growth phase, and (B) stationary phase. The blue lines represent the coverage depth processed using a median filter with 100 nt for both the stride and window length. The black arrows indicate the position of cdc6 in the genome. The lines inside the circular plot express the magnitude of the weighted pPTR for H. volcanii in the (C) exponential growth phase and (D) stationary phase. The circles represent positions of replication origins, and the crosses represent the positions of the centers of gravity of the replication origins. The lines inside the circles represent the relative magnitudes of the weighted pPTR.

Discussion

Here, we introduced a generative statistical model of coverage depth based on circular statistics and evaluated the estimated growth dynamics, replication trend, and differences in wPTR among multiple origins. In directional statistics, the simplest approach to expressing angular bias may be the use of the MRL. Although the MRL of the coverage depth was correlated with the experimental growth rate in the culture datasets, it is not as robust as estimates obtained via statistical models. This statistic can be easily calculated even with poor computational resources, but is not suitable for metagenomic datasets. Our proposed method was as accurate as the previous methods when compared with the experimental growth rates, and furthermore, it was robust against random mutations in the reference sequence and decreases in the coverage depth. Conversely, it was sensitive to a decrease in coverage depth due to mutations concentrated in a specific direction as well as to an increase in coverage depth due to conserved regions. In future research, it is expected that the rapid increase or decrease in coverage depth will be modeled to more accurately estimate the dynamics of the coverage depth. The simplest approach is not to use coverage depth in regions that are expected to be ineligible, as has been done in previous studies. However, this filtering approach alone does not provide a reasonable estimation for the proposed model as it also uses the absence of observation for parameter estimation. If a valid region [a, b] can be assumed, ineligible regions could be excluded by normalizing the likelihood function to satisfy ∫abpθdθ=1. In applying the proposed method to the coverage depth obtained from the metagenomic sequence, the average coverage depth and random sampling properties must be examined, as was done here. Although we did not utilize it in this study, one of the advantages of a GLM is its ability to incorporate covariate effects into the model. If one wants to evaluate the relationship between the covariates x and pPTR, it is suitable to use a link function for the concentration parameter. For example, when the von Mises distribution model is used, let β be the coefficient of the covariates; then κ = exp(β0 + β1x1 + ⋯) satisfies the requirement, i.e., κ > 0. For a wrapped Cauchy distribution, the inverse logit function is appropriate to satisfy 0 < ρ < 1.

We demonstrated the generation of artificial coverage depths using our statistical model. It was confirmed that the shape after sorting in ascending order was similar to the experimentally obtained replication profiles, which were presented previously (Brown et al., 2016). This shape did not depend on the type of circular distribution. These results demonstrate that the distorted shape could be generated not only by prophage sequences, strain variations, and highly conserved regions but also by the randomness of observation (DNA sequencing). When we evaluated this shape using actual data, we observed its appearance even in partial sequences that did not include these regions. This finding suggested that the shape is attributable not only to specific regions but also to the variance in observations, which is modeled by a multinomial distribution in our model. Filtering these parts out undoubtedly reduces the noise in the coverage depth.

In the evaluation using the in vivo dataset, we successfully confirmed consistency with previous studies. The species in the Bifidobacterium genus showed growth diversity when our method was evaluated using fecal WGS from infants and their mothers. Previous studies have revealed that Bifidobacterium adolescentis is abundant in adults and Bifidobacterium breve is abundant in infants (Turroni et al., 2012; Ruiz-Moyano et al., 2013; Kato et al., 2017); this trend was also reflected by the growth estimates. Although this finding was not reproduced by PTRC, previous studies have indicated that B. breve grows faster than other Bifidobacterium species in formula milk (Dubey & Mistry, 1996) and human breast milk (Turroni et al., 2011). Since not all of the infants in the dataset had been weaned at the time of the study (Bäckhed et al., 2015), it is suggested that our method appropriately interpreted the dynamics.

When we compared the non-extended directional distributions for the replication trends, the Jones-Pewsey distribution exhibited the best fitness. This result implies that the additional parameter could contribute to the coverage depth dynamics that had been overlooked. The additional shape parameter implied that more reads were concentrated around the replication origin in the early stage of the exponential growth phase, except for L. gasseri in an anaerobic culture. We additionally applied the InvSE model to evaluate this phenomenon based on another quantification; this model reproduced the trend obtained using the Jones-Pewsey distribution model. We provide two possible explanations for the above phenomenon. The first is the effect of multiple replication forks. As the cell division phase is shorter than the genome replication phase in bacteria, the genome begins replication before finishing the current replication origin (Cooper & Helmstetter, 1968; Bremer & Churchward, 1977; Yoshikawa & Wake, 1993; Wallden et al., 2016), allowing multiple rounds of replication to occur around the replication origin while rapid replication is occurring. Emiola and Oh also discussed the effect of multiple fork replication on the coverage depth (Emiola & Oh, 2018). The second hypothesis is that, as the entire chromosome is not affected at the start of DNA replication, some DNA appears only around the replication origin. However, this concept does not explain the generation time of bacteria. Under laboratory conditions, DNA replication of E. coli is reportedly completed within approximately 30 min (Helmstetter & Cooper, 1968). If the second hypothesis was valid, the additional coverage depth concentration around the origin should be finished within 30 min from the beginning of the culture. However, the degree of density remained low for an hour in our study. The trend observed in the anaerobic cultured L. gasseri was the opposite of what was seen in the others; however, it is worth noting that L. gasseri required 90–120 min of adjustment to have a sufficient correlation between the experimental growth rate and estimated growth dynamics. This suggests that both the activation of DNA replication and cell division are required to decrease the degree of density. Accordingly, we inferred that the degree of density and peakedness may indicate the activity of multiple replication forks. In contrast, the skewness parameters in E. coli and L. gasseri did not change dynamically during the experimental duration. Additionally, we confirmed the presence of a strong correlation between the skewness of the Watson and Crick strands, implying that the amount of DNA remains symmetric between the Watson and Crick strands as well as between the leading and lagging strands.

In addition to the application to microbes with a single replication origin, we extended the model’s application to microbes with multiple origins in a single chromosome. One interesting finding was the difference in wPTR among multiple origins. From the relationship between the intermediate position of the origins and the split position of the chromosome sequence based on wPTR, the efficiency, in terms of the activity of the origins, was quantitatively confirmed. If only a single replication origin was active in a chromosome, considerable time could be required for whole-genome replication, which would be a disadvantage for survival. By properly activating the origins at a distance, replication may be efficiently completed. However, our results indicated that not all replication origins exhibit similar activity. There are various characteristics that cause the activity to differ, such as (a)synchronous initiation (Lundgren et al., 2004), replication fork speed (Elshenawy et al., 2015), and so on. Therefore, the mechanism underlying the differences observed for each replication origin must be clarified, and the characteristics of neighboring genes must be investigated.

The current study was affected by certain limitations. First, the proposed method requires circular genome sequences for accurate estimation. As several methods involving contig or scaffold-level sequences have already been proposed for estimating the quasi-growth of bacteria (Brown et al., 2016; Emiola & Oh, 2018; Gao & Li, 2018), it is recommended that these methods be properly used depending on the accuracy requirements. It is difficult to detect trends in the amount of DNA other than the coverage depth bias or to estimate the bias in chromosomes with multiple replication origins using these methods. We consider our method to be appropriate for data analyses related to detailed replication profiles.

Second, the taxonomic resolution is limited to the species level in our method, on account of the first limitation. When the growth estimates of a reference strain were calculated using metagenome samples containing different but closely related strains, their growth dynamics were found to be different, indicating that the pPTR distributions may be mixed. This difficulty regarding the taxonomic resolution has yet to be solved via growth rate estimation, which may give rise to major challenges in environments such as soil, wherein many closely related species are contained because of empty niches and/or microstructures (Dumbrell et al., 2010; Thompson et al., 2017). However, this challenge may be less serious in environments devoid of close relatives on account of the filling of niche space and/or strong selective pressure. The human intestine likely corresponds to the latter case (Jeraldo et al., 2012; Li & Ma, 2016; Thompson et al., 2017) but may shift to the former case in situations in which the population is being reconstructed because of an environmental change (Langenheder & Szekely, 2011). This resolution problem may be solved by constructing pan-genome sequences from metagenomic reads and allocating coverage depth appropriately. Third, the evaluation scope of the extended model is limited. Although we evaluated and eliminated the possibility of overfitting in our dataset, we cannot deny the possibility that the dynamics of the peakedness and stability of the skewness around the origin are specific to the three strains we used. External validations are expected to confirm the variability of the peakedness or stability of the skewness over the growth phase. Finally, in our method as well as all currently proposed methods for estimating bacterial growth, the estimate itself is only a proxy of the growth rate. Theoretically, (p)PTR for a taxon t in a sample s is represented by (p)PTRs,t = 2Ct∕τs,t, where Ct is the replication period and τs,t is the doubling time (Cooper & Helmstetter, 1968; Bremer & Churchward, 1977; Korem et al., 2015). Our interest is in the doubling time, but the estimate is also influenced by the replication period. This period may vary from species to species depending on the genome size and other factors. Therefore, it is not appropriate to compare estimates between species. It is necessary to analyze the effects of the replication period C and to propose a method that yields a doubling time that is comparable between species (Gibson et al., 2018).

Conclusions

We developed a probabilistic model based on circular statistics to model the coverage depth behavior in DNA replication using WGS reads. This method was demonstrated to be robust for a small number of reads (≥0.01 ×). The probabilistic PTR from our model demonstrated a significant correlation with the experimental growth rates in the culture dataset. In addition to facilitating quantification of the ratio differences, this method enables detailed measurement of DNA quantity changes by using circular distributions in the model. Moreover, by combining multiple distributions, it became possible to estimate the growth of organisms with multiple replication origins, such as archaea. Therefore, this method further extends the applicability of growth estimation from fragmented reads. We expect that the growth estimation method presented herein will help elucidate factors that have not yet been observed in studies of microbiome formation.

Supplemental Information

Figure S1 Errors of unnormalized and normalized von Mises distribution-based models

We fitted the unnormalized and normalized von Mises models to a simulation dataset to evaluate the error with respect to the true value. The vertical axis represents the Kullback-Leibler divergence with respect to the true distribution (kld), and horizontal axes indicate the parameters. The green bars indicate the unnormalized von Mises distribution-based model, and the pink bars represent the normalized von Mises distribution-based model when the (A) location parameter was changed, (B) the concentration parameter was changed, (C) the discrete length was changed, and (D) and average coverage depth were changed.

Click here for additional data file.

Figure S2 Correlation between growth estimates and experimentally obtained growth rates

We evaluated the accuracy of the method using experimental growth rates. (A) Method of computing the correlation between the growth estimates from the WGS and the growth rates experimentally obtained from CFU/ml in the L. gasseri OX culture. Considering the delay of cell division due to DNA replication, the growth estimates were shifted before computing the correlation coefficients. The yellow circles represent the existence of data, and the arrows represent the inheritance of values. (B) Growth dynamics according to the experimental growth rates and estimates from the von Mises distribution model. The parameter of the median filter was 100 bp in both cases. The Pearson correlation was calculated separately with shifts for the L. gasseri OX culture (60 min) and NOX culture (120 min). (C) Comparison of the accuracy of this method with those of previous methods. The evaluation was conducted using the experimental growth rate computed according to Korem et al. and (D) using a differential approach to investigate the short-time dynamics. (E) Evaluation of the correlation using a mixed community dataset containing seven species. A high correlation is observable for L. gasseri (r = 0.76 ± 0.04) over four biological replicates. Each panel exhibits a trend in the growth estimate and experimental growth rate in the single replicate from A to D.

Click here for additional data file.

Figure S3 Effect of median filter

We calibrated the window and stride lengths in the median filter to evaluate their effects on the PTR estimation. The Pearson correlation coefficients between the estimated PTRs and experimental growth rate were obtained using different median filter parameters. We compared the PTRs using MAP estimation. Except for the E. faecalis dataset, a high correlation with the experimental growth rate (≤100 nt for both parameters) was maintained. The same files were used to compute the PTR for the same median filter parameter set in multiple models. The sample size n represents the time point used to compute the correlation coefficient.

Click here for additional data file.

Figure S4 Smoothing effects of different window sizes

Effect of changing the window length in the median filter on the coverage depth. The y-axis represents the ratio of the coefficient of variance after the application of a median filter prior to filtering.

Click here for additional data file.

Figure S5 PTR error caused by coverage depth decrease at the edges of a sequence

To evaluate the effect of the coverage decreasing at the edge of a sequence, we compared (A) the PTR from the raw sequence with that of the hang-over reference sequence and (B) the PTR obtained using Bowtie2 with that of vg. Both comparative evaluations were performed using a von Mises distribution-based model and the L. gasseri WGSs from Korem et al.

Click here for additional data file.

Figure S6 Computational resources required to estimate parameters and statistics

The figures on the left represent the required CPU time and maximum memory usage when the stride length of the median filter was varied, and the window size was fixed to 100 nt. All target data were obtained from NOX cultured L. gasseri WGS depths. We could not estimate the parameters of the Jones-Pewsey distribution based-model when a memory of less than 512 GB was used at a stride length of 1 nt. The figures on the right represent the required CPU time and maximum memory usage with varying coverage depth. The median filter used a window and stride length of 100 nt. All computations were performed via MAP estimation.

Click here for additional data file.

Figure S7 Error of growth estimates in mutated or contaminated datasets

The robustness of the estimates was evaluated using artificial datasets. (A) The error rate for each average coverage depth was calculated with respect to the full coverage depth. Only the L. gasseri WGSs that had greater than 20 × coverage were used. The reference genome sequence was modified by (B) nucleotide level mutation, (C) block-level mutation on a random position, or (D) block-level mutation on a single specific region. The WGS was contaminated by artificial reads from the reference genome sequence at (E) a single position or (F) multiple positions so that the peak depth could be generated. (G) Even when we decreased the sample size, the effect was not substantial compared to the full set of samples. The horizontal bar represents the 15% threshold of the error rate defined previously (Brown et al., 2016). The black crosses on the bar indicate that the method was unavailable for mutated genome sequences. The WGS datasets were obtained from previous studies (Korem et al., 2015; Franzosa et al., 2018). The proposed models and statistics are placed in the black rectangle within the legend.

Click here for additional data file.

Figure S8 Evaluation of maximum deletion size in IBD cohort dataset

The size of the largest deletion was estimated from the coverage depth obtained when mapping the fecal metagenomic sequence of a human IBD patient cohort study (Franzosa et al., 2018) to a complete genome sequence database.

Click here for additional data file.

Figure S9 Relationship between noise coverage fraction and zero coverage depth fraction

To infer the noise fraction in the coverage depth, we investigated the relationship with the zero-coverage depth fraction. (A) Correlation of the noise fraction with the error of our estimates in samples with more than 5.0 × average coverage. The dataset with multiple artificial peaked noise coverage was used. Evaluation of the Lander-Waterman theory in E. coli, E. faecalis, and L. gasseri datasets for (B) raw coverage depth datasets and (C) datasets after applying the moving median filter. We used a theoretical model assuming uniform probability. The variation ranges were visualized by 30%, 50%, and 70% of the log-scale score, respectively. Mean fold change between the log zero coverage fraction and theoretical score using (D) all noise-contaminated samples and (E) samples with less than 5.0 × average coverage. (F) Correlation between the noise coverage fraction and the fold change of the log zero coverage fraction in samples with less than 5.0 × average coverage. (G) Sample distribution with respect to the log zero coverage fraction. (H) ROC curve for estimation error determination when the threshold of the log zero coverage fraction was changed (AUC: 0.81). At the threshold for maximizing the F1 score, the accuracy was 0.77, the precision 0.78, and the recall 0.81.

Click here for additional data file.

Figure S10 Relationship between replication and random sampling

Based on the theory of Lander and Waterman, we investigated the relationship between the zero-coverage fraction and average coverage. We set the simulated sequence size to 10,000 nt. The observation probability at each nucleotide position was determined based on the von Mises distribution. (A) The y-axis represents the fold change of the log-transformed zero coverage fraction to that assuming uniform probability. The greater the PTR, the greater the deviation from the model assuming uniformity. This difference increases particularly when the average coverage is large. (B) Theoretical zero coverage fraction based on the Lander-Waterman theory when the pPTR was changed. The variation ranges in a model assuming uniform probability are shown at 30%, 50%, and 70% of the log-scale scores. (C) Log-scale score. (D) Fitting of the exponential-power function to the theoretical scores. The parameter of the function was determined via the least-squares method using Scipy.

Click here for additional data file.

Figure S11 Evaluation of median filter in metagenomic datasets

In the real metagenomic dataset, noise smoothing of the coverage depth by the median filter was evaluated based on the coefficient of variation. The data were obtained from human intestinal metagenomic sequences related to (A) infants (Bäckhed, et al., 2015), (B) IBD (Franzosa et al., 2018), and (C) colorectal cancer (Yu, et al., 2015).

Click here for additional data file.

Figure S12 pPTRs of mice infected Citrobacter rodentium

pPTRs were estimated using the von Mises distribution-based model. As in the work by Korem et al., significant differences were detected between the wild type and mutants on days 6–9 (FDR-corrected p-value: 8.72 ×10−5) and between days 1–5 and 6–9 for the wild type (FDR-corrected p-value: 1.04 × 10−5).

Click here for additional data file.

Figure S13 Growth estimate distribution of Bifidobacterium species in feces microbiome of neonates and their mothers

Estimated growth of B. adolescentis and B. breve in fecal samples from neonates and their mothers obtained from the (A) von Mises distribution model and (B) PTRC. This cohort was composed of 100 neonates and their mothers. Three fecal samples were obtained from the neonates just after birth, four months after birth, and one year after birth. The feces of the mothers were sampled just after birth. Bars with single stars indicate significant differences in the growth estimates according to Welch’s t-test (p-value ≤ 0.05), and the bars with double stars indicate significant differences after FDR correction (FDR-corrected p-value ≤0.05). For the correction, we followed the Benjamini-Hochberg procedure.

Click here for additional data file.

Figure S14 Degrees of concentration density, peakedness, and skewness around the replication origin

Degrees of density, peakedness, and skewness of replication according to the (A) Jones-Pewsey, (B) InvSE von Mises, and (C) InvMIAE von Mises distribution-based models. Except for the Jones-Pewsey distribution model for E. faecalis and L. gasseri, the parameters were estimated using the MAP estimation algorithm, while those of the other models were estimated using the MCMC algorithm. The estimates for all metagenomic samples obtained from the same species were obtained by sharing the location parameters regardless of the cultural state. The WGS datasets were obtained from Korem et al.

Click here for additional data file.

Figure S15 Correlation between shape and peakedness parameters

Scatter plot of the degree of concentration (psi) and peakedness (lambda). We calculated psi for E. coli and lambda using the MAP estimation, and calculated psi for E. faecalis and L. gasseri using the MCMC algorithm. The parameters were analyzed regardless of cultural status.

Click here for additional data file.

Figure S16 Evaluation of parameter optimization of asymmetry extended model with simulation data including pulse-shaped noise

To evaluate the detection power in Pewsey’s asymmetry test and the invMIAE model, we computed the p-values of the test and skewness parameter of the model using the simulated dataset. Random numbers were iteratively generated following the (A) symmetric von Mises (nu = 0), and (B) asymmetric von Mises distribution (nu = 0.5) and multinomial distributions 50 times for each pulse strength. Pulse noise was added to the position that nearest to a quarter of the unit circle. The log-likelihood difference between the pulse-less true model and the estimated model is shown in the bottom row in each figure. The robustness of the model was also evaluated using (C) artificial peak noise and (D) an artificial mutation dataset.

Click here for additional data file.

Figure S17 Coverage depth and probability density of InvMIAE model for E. faecalis

The pink square indicates the suspectable region that made the model skew.

Click here for additional data file.

Figure S18 Skewness comparison between Watson and Crick strands

The parameters for each species were estimated by performing separate optimization trials. The correlation coefficient and p-value were calculated using all measurements.

Click here for additional data file.

Figure S19 Multiple replication origins in Sulfolobus solfataricus

Growth rate estimates were performed on S. solfataricus using data sets obtained from Payne, et al. Subplots depict datasets from (A, G) SARC-B, (B, H) SARC-C, (C, I) SULA, (D, J) SULG, (E, K) SARC-H, and (F, L) SARC-I. Robustness of the wPTR and mwPTR estimates was evaluated using artificial datasets with (M) decreased coverage depth, (N) point mutation applied, (O) 5,000 nt block mutation applied, and (P) peaked noise inserted.

Click here for additional data file.

Table S1 Software versions

Click here for additional data file.

Table S2 Genome and metagenome sequences used to verify the parameter set of the moving median filter

Click here for additional data file.

Table S3 Chromosome sequences in complete genome sequence database

Click here for additional data file.

Tabale S4 Odds ratio of the sequence feature frequency in the skewed region at both ends

Click here for additional data file.

Table S5 Fitness of the model to coverage depth datasets by WAIC

Bold style had the lowest score (the best fitness).

Click here for additional data file.

Table S6 Deviation between actual zero coverage fraction to Lander-Waterman theory

For the expected fraction, the exponential model based on the Lander-Waterman theory assuming uniform probability was used.

Click here for additional data file.

Table S7 Results of two comparison test of growth rate in infant cohort

We used Welch’s t-test for the p-value and the Benjamini-Hochberg procedure for FDR correction. p-values less than 0.05 appear in red.

Click here for additional data file.

Table S8 Parameter settings for the models

Correspondence between circular distributions and types of parameters.

Click here for additional data file.

Table S9 Information criteria of the proposed models obtained using the data of Korem et al

To compare the models based on their prediction performance, both AIC and BIC were computed. The lower the information criterion, the better the model fit of the data. For each dataset and criterion, the values in bold are the lowest values in the vanilla models and those in red are the lowest ones in all of the models.

Click here for additional data file.

Table S10 Information criteria used to predict the number of replication origins in the genome sequence

The bold values are the lowest (best fitness) scores in the set of genome sequences and the information criteria.

Click here for additional data file.

Supplemental Information 1 Supplementary text

Click here for additional data file.

We thank Ken Kurokawa, Hiroshi Mori, and Koichi Higashi for discussions regarding the research design during the early stages of this work. We thank Yuya Nakamura for discussions regarding the statistical modeling experiment and design of the linear cardioid model. We thank Darzi Youssef for his review of the in vivo dataset section. We thank Anton Björk for his essential advice related to the skewable asymmetric extended model. We thank Thorsten Allers for providing the H. volcanii genome sequence. We are grateful to the reviewers for their numerous helpful comments, which helped us to improve this report.

Additional Information and Declarations

Competing Interests

Author Contributions

Data Availability

Takuji Yamada is a founder of, and a shareholder in, Metabologenomics, Inc. He currently serves as the chief technical officer of the company.

Shinya Suzuki conceived and designed the experiments, performed the experiments, analyzed the data, prepared figures and/or tables, authored or reviewed drafts of the paper, and approved the final draft.

Takuji Yamada conceived and designed the experiments, authored or reviewed drafts of the paper, and approved the final draft.

The following information was supplied regarding data availability:

Software:

We implemented the statistical model using Stan (Carpenter et al., 2017). Wrapped by Python scripts, this model is available for use in the command-line environment. This package also contains a moving median filter, visualizer, statistics profiler based on directional statistics, information criterion calculator with estimated results, asymmetric test calculator using Pewsey’s method and other utilities required to analyze the coverage depth over replicon. Other software versions are summarized in Table S1. Our package for growth estimation is available from https://github.com/TaskeHAMANO/SPHERE. This software was implemented using Python3 (≥3.6) and Stan. The wrapper software used in this study for PTRC, DEMIC, and GRiD is available from https://github.com/TaskeHAMANO/PTRC-in-cwl, https://github.com/TaskeHAMANO/DEMIC-in-cwl, and https://github.com/TaskeHAMANO/GRiD-in-cwl, respectively. This software is distributed under BSD-3-Clause license. The wrapped software of msbar in EMBOSS is available from https://github.com/TaskeHAMANO/msbar-in-cwl. This software is distributed under GPL-3.0 license. These wrapper scripts were implemented using Common Workflow Language (CWL) v1.1. These scripts have been tested on Linux and macOS.

Availability of data and material:

The WGS data of time-series-cultured E. coli, E. faecalis, L. gasseri, S. solfataricus, and H. volcanii are available from BioProject (PRJEB9718, PRJNA250819, PRJNA250820, PRJNA250827, PRJNA346830, PRJNA250832, PRJNA250833, and PRJNA422812). The genome sequences of E. coli NMC3722, E. faecalis ATCC 29212, L. gasseri ATCC33323, S. solfataricus SULA, SARC-B, SARC-C, USLG, SARC-H, SARC-I, and H. volcanii DS2 are available from GenBank and RefSeq (CP011495, CP008816, NC_008530, CP011057, CP011055, CP011056, CP033235, CP033236, CP033237, and NC_013967). The genome sequence of H. volcanii H26 was modified from DS2 as described previously (Hawkins et al., 2013). The genome and metagenome sequences used in the cohort studies analysis are listed in Table S2. The final chromosome sequences we have used to construct the genome sequence database are listed in Table S3.

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
