# Peer review of "Probabilistic model based on circular statistics for quantifying coverage depth dynamics originating from DNA replication"

_PeerJ, doi:10.7717/peerj.8722_

## Round 0.1 · original submission · Major Revisions

The reviewers commented on a number of grammatical errors and scientific inaccuracies in the manuscript. In addition, there is a need to add references to existing literature on the topic. Both reviewers have made a detailed and very helpful list of recommended changes that I encourage you do carefully address. Please avoid overstatement of significance of your results.

·

Basic reporting

Literature references up to date.
Figure adhere to a high quality.
Manuscript would benefit from restructuring, see General comments for suggestions.

Experimental design

Authors propose a tool based on circular statistics to infer information on growth of prokaryotic population from genome sequencing coverage. Analysis well planned, rigorous investigation performed.

Validity of the findings

Few aspects need to be clarified, see for details General comments.

Additional comments

Major remarks:

Methods proposed by (Koremet al. 2015) and (Brownet al. 2016) have been criticised in (Gibson et al. 2018): “In principle, these measures of cells performing DNA replication could be used to estimate the doubling time of bacteria in the wild. However, it is unclear how or whether the methods can be calibrated.” An example of such calibration was done in (Forsyth et. al 2018), figure 3D, where a linear model was constructed as growth_rate = 1.678*PTR - 1.772.
Therefore I think that in the manuscript “pPTR” should not be substituted by “growth rate”. Say on line 597: “estimating a correlation between computational and experimental growth rates as per a previous study (Korem et al., 2015)” - This should be “ between pPTR and experimental growth rate”. If You look how (Korem et al., 2015) phrases it, it’s “PTRs were correlated with the measured growth rate, preceding it by 30 min” and “1/log2(PTR) [] correlated with measured bacterial generation time, confirming PTR as its proxy, even when the replication time is unknown.”, as well as “The relationship between PTR and growth rate extends to other commensal strains, as we found that PTR and temporal measured growth rate were significantly correlated ...”
The same applies to lines 603, 605, 607, 614 etc.

Gibson B, Wilson DJ, Feil E,Eyre-Walker A. 2018 The distribution of bacterial doubling times in the wild.Proc. R. Soc. B285: 20180789.http://dx.doi.org/10.1098/rspb.2018.0789
Forsyth VS, Armbruster CE, Smith SN, Pirani A, Springman AC, Walters MS, Nielubowicz GR, Himpsl SD, Snitkin ES, Mobley HLT. 2018. Rapid growth of uropathogenic Escherichia coli during human urinary tract infection. mBio 9:e00186-18. https://doi.org/10 .1128/mBio.00186-18

Line 613: “Secondly, we tuned the parameters concerned. The window and stride size of the median filter were optimized…” & from supplementary material: “The purpose of the filter is to reduce the variance in the coverage depth and outliers. As such, if filtering is performed, ideally, the average depth will be maintained and the variance will decrease.” I don’t see why You need median filtering when You are fitting coverage depth to a function? Zero depth shouldn’t be an issue, as function is fitted to all data and keeping in mind a limited number of parameters to estimate, it is mostly influenced by high coverage areas. As You show in Supplementary fig. 6, 10, your fitting results are sensitive to the choice of window length.

It is nice to see the theory extended to a multi-origin case. However, I would argue that the maximum of wPTR is more appropriate index as a proxy for growth rate.
“These scores are based on the hypothesis that replication stops if the replisome is faced with another replisome (Wendel, Courcelle & Courcelle, 2014).” - this is a reference on a single origin bacteria.
Synchronous initiation of replication origins has been reported in Sulfolobus species (Lundgren et al., 2004), but not in H.volcaniil (Hawkins et al., 2013). Furthermore, the difference in height of fitted mixture components indicates difference in initiation of replication (figure 4 a & Supplementary fig. 21 a-f).

Other remarks:

Introduction.
Although lines 166-170, 363-367, Figure 2 etc talk about PTRC, DEMIC, bPTR, iRep, and GRiD, these are not properly introduced. Add a paragraph to describe previously used indexes (not just PTR). Make it clear that You are benchmarking Your method against existing ones and that You have made a package for Your method.

Materials & Methods.
I can’t find if coverage data have been binned into 10kbp regions as in previous studies? If yes, then methods need to be revised, as i is not nucleotide, but a bin on the chromosome. By the look of Figure 3&4, these don’t seem to be at 1 nt resolution.

Move subsections “Software” and “Availability of data and material” to the end of Materials and Methods. Move lines 166-168 to “Software”, i.e that SPHERE is Your package for computing directional stats from coverage depth data.

Start “Materials and methods” with subsection “Circular distributions and statistics”, as this is the main aspect of the manuscript. Put all probability density functions into a table with the following columns: (i) the name of the distribution, as used in Figure 2; names are missing from lines 179-183, but then they are used on lines 204, 255 etc and results section; (ii) formula of probability density function; (iii) analytical expression for PTR (given on lines 199-203), or numerical method used for calculation; (vi) parameters defining probability density function; and (v) and reference for this probability density function.

Line 222: “Fitness of the model was improved..” - was this done for fitting to the data to models? Then lines 222-225 should be moved to “Parameter estimation”.

I would suggest moving lines 516-578, i.e subsection “Statistical model to estimate replication rate” to “Materials and methods”, and adding it after “Parameter estimation”.

Line 520: “We fit a generalized linear model (GLM)..” This is confusing - i thought You’re fitting probability density functions defined in subsection “Circular distributions and statistics”?

Line 563: “This score represents the latent bias of probability for the position at which a nucleotide is observed around the replication origin.” Can you please clarify? As p_max(theta) and p_min(theta) are point estimates of minimum and maximum of probability density function, why there should be “latent bias” around replication origin?

Line 566: .”Following the model, the coverage depth at a given position can be modeled by the binomial distribution...” Which model?

Move lines 570-578 to supplementary materials.

Lines 579-593 - make this into a new subsection “Creating an artificial coverage depth” or similar and keep this in “Results” section. I really like Supplementary fig. 3, would be nice to move it into the main text. But provide parameter values for all distribution in the figure legend.

Line 674: “The relatively weak negative correlation between relative abundance and growth rate, became stronger when the dataset was transformed to log-scale (Supplementary fig. 14).” Can state values of correlation (i.e. -0.1172 & -0.215) rather than “ became stronger”.

Line 787: “Although it was more fragile than when..” What do you mean by “fragile”?

Line 794: “replication balance among multiple origins” - i don’t think that this has been estimated or interpreted in biological context. As well on line 946: “One interesting finding was the balance of replication activity among multiple origins.”

Figure 1: I don’t think that comments “Previous methods of in situ growth rate estimation used linear trend regression in the log-transformed Euclidean system.” or “Similar to previous methods” are necessary in the legend.
Figure 2: “PTR error rates” - how these were calculated for iRep, GRiD & DEMIC indexes? Can You plot raw values with extra bars plotted for “Full coverage”, i.e Estimte_reference and Estimate_modified as in equation (3)?
Figure 4: can You make ranges smaller, so that mixture distribution overlays coverage depth, in a similar as it is shown in Figure 3?

Minor corrections:

Line 87: “ piecewise polynomial linear regression” - it’s either polynomial or linear. In the case of Korem et al., 2015, it is a piecewise linear function (page 7, supplementary materials).

Line: 141: “Mixing directional distribution” - Using a mixture of directional distributions

Line 178: Define what is theta and T.

Line 242: “skewness or peakedness parameter” - is that the same parameter - lambda?

Line 751: “by mixing circular distributions” -> a mixture of circular distributions

Line 771: “we determined the number of components in both datasets as 3 because it yielded the best score on average (Supplementary table 9)” - there’s some mixed-up with tables, as “peerj-42075-Supplementary_table_9.xlsx” contains “Supplementary table 8: Two comparison test result of growth rate in three metagenomic cohort. We used Welch's t-test for the p-value, and Benjamini-Hochberg procedure for FDR correction. “

Reviewer 2 ·

Basic reporting

1. There are multiple instances in the paper where the grammar and structure of the sentences is not good and makes it hard to follow.

2. The introduction is argumentative rather than informative and could be further shortened and clarified.

3. The discussion need not repeat each and every result in the paper. It is way too long in my opinion.

4. L99-104 – prediction of the origin of replication was actually done in Korem et al, using their statistical model (see their Fig. S9).

5. L74-76 – Where is the limitation? What is the issue with applicability via single metagenomic samples?

6. I suggest to put Fig. 2A with bPTR in the supplementary and focus on the others for readability. Also, in Fig. 2B, it’s impossible to understand what are the methods compared.

7. Fig. S9B – there is no point in taking the X-axis up to 50%. It’s not even the same microbe at this point. The fact that pPTR even output something at these rates is alarming and indicates that there isn’t a good control for the actual presence of the species in the data.

8. L273-277 - What is the procedure to determine if top 1% should be filtered?

9. L284 – what correction procedure is being referred to?

10. L362-363 – It’s not clear where that error rate was used and what it was defined for.

11. L3375-383 – It is not clear what is “randomized direction” and “single direction”.

12. L402-403 – what are “points”?

13. L485-487 – It is not clear what “filtering out” procedure is being referred to here.

14. L532-534 – This is the only place in the results where the MRL is mentioned. The authors treat it as if it is another method (e.g., in the discussion) but it is never used in the manuscript.

15. L539-L543 – Not clear how this sentence fits in here. Are the authors trying to justify non-linearity of the distributions? It’s not clear.

16. Legend of Fig. S4 is insufficient to understand panels c-e.

17. L595-L612 – the authors should explain the experiments whose data they are using even if these were described in previous studies.

18. What is “ptr” on the legend of Fig. S8?

19. Fig. S10B – how many points are at the (1,4) and (4,1) coordinates? Could be multiple overlaying.

20. L646-L656 – Don’t assume that the readers know that it’s the same dataset with different sample size analyzed in both this and Korem et al.

21. L674-L676 – Where does this result fall within the narrative of the analysis?

22. L78 – DNA increases considerably…via DNA deep sequencing? I would say the increase is biological.

23. L362 – It’s not clear what mutations are being referred to.

24. L579-580 – The sentence is not in proper English.

25. L663-L666 – What does “confirm the trends” mean? What trends? “This trend was reproduced in the estimates” – I don’t understand what this means.

Experimental design

1. The authors assess the robustness of their method on a very limited and clean datasets of monocultures. This is valid independently, but there is no point in comparison to different methods that were optimized to work on metagenomic datasets, perhaps making decisions that improve robustness in “dirty” metagenomic data in favor of accuracy in monoculture data.
This subject came up in previous review and the authors claimed they need a ground truth measurement. This is true for assessing accuracy, but absolutely not true for robustness. The point in robustness is to see whether the method is sensitive to low coverage, mutations, increased coverage, etc., regardless of how accurate it is (accuracy is assessed separately). In fact, the authors are not using the ground truth in the robustness analysis at all. The legend of Fig. 2 clearly states that they are comparing PTRs on subsampled data to PTR on full data - ground truth doesn't play here. This was done in Korem et al., for example, where all the robustness analyses are done on a metagenomic cohort.
In my opinion, the authors cannot claim that their method is more robust than other methods unless they perform this comparison on metagenomic data.

2. The authors make arbitrary decisions when selecting microbes for the analysis of metagenomic datasets (e.g., abundance larger than >0.5%, high coverage), that are unacceptable. There is also no point in selecting taxa with already significant results at the relative abundance level for further analysis, esp. as Korem et al showed that relative abundances are not correlated with growth rates and that they found multiple significant correlations between PTRs and host phenotypes in bacteria whose relative abundances were not correlated with the same phenotype. The justification the authors provide in L476-478 is not acceptable: if the method presented here is so sensitive that it cannot distinguish taxa that are really present in the sample from those that are not, it should either be amended, or the authors should not claim it is useful for metagenomic analyses. Finally, these arbitrary decisions result in very small reference datasets – the authors basically compare 10-20 taxa – while previous methods work with thousands.

3. Fig. S4, c-d: (i) it is not clear what data was used here. (ii) From the methods, it seems that the authors used the abundances when calculating the experimentally derived growth rates. The log abundances should be used. See e.g. Hall et al. (2013) Mol. Biol. Evol. 31(1):232–238. (iii) The authors claim that their correlation was better (L607-L609), was this statistically significant?

4. L609-612 – there are no experimental growth rates in this experiment. It is impossible to calculate growth rates from relative abundance due to compositionality of the relative abundances.

5. L539-L543 - What is the point of Fig. S1? How does it relate to the sentence? Do the authors claim they can see non linear trends there? Can they prove it? Seems like a reproduction of Fig. 1C from Korem et al.

6. Why is FDR correction necessary for Fig. S11? (L645) Seems like just one test?

Validity of the findings

1. The authors should either prove that their method works well on metagenomics datasets or limit their claims to monocultures. The authors fail to show that their method works on metagenomics on multiple accounts:
a. The robustness analysis is done only on monocultures.
b. pPTR seems incredibly sensitive to mutations at specific locations, that are typical in metagenomic datasets (Fig. S9C, S9E).
c. The analysis the authors do perform on metagenomic cohorts is handicapped by arbitrary selection of species to work on, resulting in a very small reference datasets (2 orders of magnitude vs. previous methods). The authors even justify that by the deficiency of their method to work in this setting (L476-478).
d. The analysis the authors perform on these metagenomic datasets is not compared to any other method. I read the authors essay about this in the rebuttal. While 80 vs 100 significant association is perhaps not a good argument to make towards the superiority of any method, if one method discovers multiple significant associations and another recovers just one or two then this is a different situation.

2. The characterization of Korem et al in L85-93 is inaccurate. Korem et al limited the coverage due to reduction in accuracy in low coverage (see their Fig. S7). The use of “division by zero” by the authors makes it seem like some error in the code, which I assume is inaccurate and even if so not very relevant. This returns in L814-816 – there is no zero division error anywhere. The limit of 0.05x was due to accuracy. These cases are filtered out and do not result in an error. This is very clear in the manuscript of Korem et al.

3. The authors also cannot claim that their method is robust to very low coverage of .005x, if they themselves limit the applicability of their method to .1x which is higher than previous methods. Their arguments to the contrary in the rebuttal are unacceptable - the bottom line is what matters. If there method cannot work with coverage <.1x - it is not applicable for coverage up to 0.005X.

4. How can the authors say that parameter estimation was independent of sample size (L635-636)? Fig. S9F clearly shows increased error with lower sample size!

5. L647-652, L850-L854 – PTRs cannot be used to compare between different bacteria. This appeared in previous peer-review which the authors reject for not good reason. PTR = 2^C/G (Korem et al, Cooper and Helmstetter), where G is the generation time, and C is the replication period, that is, the time it takes a microbe to completely replicate its genome. The C-period is different between different microbes. Therefore one cannot say that one microbe is growing faster (smaller G) than the other just based on PTR, without knowing C. This argument is valid only within a species where one can assume that the C-period is similar.

6. L651-652 – What exactly is consistent between the two analyses?

7. L806-809 – The comparison was done on a monoculture, while these methods were developed for metagenomics, and thus is invalid.

8. L856-L857 – this sentence is false. On a 10M 100bp reads sample and a 5Mbp genome, relative abundance of 0.1% corresponds to coverage of 0.2X, which is not low.

9. L868-L870 – this is not supported by the analyses, as mentioned above.

Additional comments

I have reviewed this manuscript before. The general notion of using a statistical model to model growth rates is an appealing one. The authors try to make a claim regarding the superiority of pPTR to previous methods and its applicability to metagenomic datasets that is not supported by their findings and analyses. They cannot claim that these analyses are just "an evaluation", and then put claims of superiority. There are multiple other inaccuracies that I point out in my review.

External reviews were received for this submission. These reviews were used by the Editor when they made their decision, and can be downloaded below.

---

## Round 0.2 · Minor Revisions

The manuscript has been re-examined by two reviewers. They have come up with useful suggestions on how to improve your paper. Please make the suggested changes and prepare the itemized list of responses to every identified issue. Hopefully it will help to bring your manuscript closed to acceptance.

·

Basic reporting

Literature references up to date.
Figure adhere to a high quality.
Article structure and language has impoved for the revised version.

Experimental design

Authors propose a tool based on circular statistics to infer
information on growth of prokaryotic population from genome
sequencing coverage. Analysis well planned, rigorous investigation
performed.

Validity of the findings

no comment

Additional comments

Minor comments:
Line 78: "bidirectional progression" -> "bidirectional progression of replication forks".
Line 80: add before "iRep used a.." a sentence "Few mathods have been proposed to quantify growth of bacteria from genomic data." or similar.
Line 94: Change "Another growth rate estimation method, which fills the gaps, could be utilized for broader.." to " A novel growth rate estimation method, which is less sensitive to coverage depth, could be..."
Line 104: What does "DNA quantity" mean?
Lines 261-264: I appreciate that the authors wrote a thorough response on my comment on multiple origin model. With regards to the following assumptions: "This model means that each replication origin is not responsible for the entire DNA sequence, but rather only for a certain region. This assumption results in the probability of regions that are not related to the origin approaching 0 or being extremely small. Hence, the probability distribution of each origin becomes very steep, increasing in the concentration parameter and unweighted PTR." But look at Supplementary figure 20 (a)-(f), all of them have one of unweighted PTRs dominating the fitted mixture.
Line 784: Define what is "the center of gravity of the replication origin position".
Line 801: "replication activity balance among multiple origins" - again, not according to supplementary figure 20.
Line 801: What do You mean "In a field"?
Line 849-850: "for DNA to be more concentrated around the replication origin" -> "for multiple rounds of replication around the replication origin".
Line 856: Plese clarify what You mean by "the increase in density around the replication origin should
be resolved".
Line 861: Please clarify: "Thus, the degree of density around the origin of replication increased when both replication and cell division were active."
Line 875: Please clarify: "If a certain single origin had strong activity".

Reviewer 2 ·

Basic reporting

The section on “Shape, peakedness, and skewness of coverage depth” is very descriptive, based on a very limited samples size, and unfocused. I recommends that it is significantly shortened, with technical details and discussion moved to the supplementary material.

I recognize that the authors have improved the English throughout the manuscript. I appreciate their efforts but there are still places in the manuscript where phrasing, grammar, etc. do not conform to professional standards. Some specific examples:

L44-45: It is not clear what this assertion of target range and measurement properties means and even if it was clear, where it was assessed.

L71 – stable isn’t the right word. Perhaps static.

L76 – after reading the authors’ response, their phrasing doesn’t say what they mean. What they want to write is something like “Meanwhile, the peak-to-trough ratio (PTR) of coverage of WGS reads mapped to reference genome sequence provides an estimate of growth using WGS reads coming from just a single sample”. The current sentence indicates something completely different and is unclear.

L77-78 – the increase is not via bidirectional progression, the increase is via replication, bidirectional progression creates the coverage signal across multiple microbes.

L78-79 – The “as well as DNA deep sequencing” part has no connection to the sentence.

L121 - The phrasing of “small number of coverage depths” is not in correct English; the “among other methods” has nothing to do with the text.

L125-126 - implies that you made sure that previous studies are reproducible: instead what you did was making sure that your method is robust to coverage and noise using data from previous studies.

L178-179.- “the low coverage depth issue” is referenced but not defined beforehand.

L278 – this sentence is unclear. What is “the package for all of the data”?

L315 – the moving median filter should be defined before it is referenced and justified.

L317-319 - "peaked noise" is not a clear term.

L320 – should be “based on a previous study”.

Experimental design

The “Statistical model to estimate replication rate” section should emphasize that the coverage is actually binned and not at the nucleotide level.

L561-565 – this procedure and its aim are completely unclear. Also: what theoretical score?

I couldn’t find the methodological details behind Fig. S3c/d

L665-667 – The text here does not correspond to the legend of S8d

Validity of the findings

L396 – this is far far away from evaluation of applicability. This is robustness pure and simple.

L669-670 – while the estimates were indeed as stable as previous methods, they were significantly less accurate (e.g. compared to ptrc, irep, grid). This should be stated and discussed.

L670-672 – Please also discuss the reduced accuracy compared to previous method. pPTR is problematic with respect to the type of variability observed in s8d/e/f and this should be clearly acknowledged.

L672-676 – It is unclear how this is addressed. The authors seems to insinuate that the simulations that they do have much more noise than real data, but
(a) one of their analyses is on monoculture sequencing of 3 well characterized laboratory strains (Text S8); this is not the same as sequencing of a metagenome. (b) their analysis deals only with absolute deletions, which is not the only source of noise in these scenarios.

Limitations with respect to metagenomic applications that the authors discuss in the rebuttal letter should be clearly presented in the discussion. This is currently not the case.

Additional comments

L81 – this is not true. iRep uses all the contigs.

L805-814 should be moved to supplementary

The authors need to make sure they have the rights to use fig s1 which is a reproduction of Korem et al


In general, this revision presents a much tighter, rigorous, and valid study with analyses that are better justified. The authors have removed statements of superiority and analyses that were not properly validated, and have addressed most of my comments.

External reviews were received for this submission. These reviews were used by the Editor when they made their decision, and can be downloaded below.

---

## Round 0.3 · Minor Revisions

Thank you for making the requested suggestions. I have read the manuscript, it is definitely improved. However, there are several places where grammar still needs polishing up. I encourage you to seek a help of a native English speaker or a technical writer/editor to proof-read the manuscript.

External reviews were received for this submission. These reviews were used by the Editor when they made their decision, and can be downloaded below.

---

## Round 0.4 · accepted · Accept

The quality of written English was improved, grammar was checked. Previous rounds of review have resolved all scientific issues. Therefore, I recommend your manuscript for acceptance.

External reviews were received for this submission. These reviews were used by the Editor when they made their decision, and can be downloaded below.